# LumiSculpt: A Consistency Lighting Control Network for Video Generation

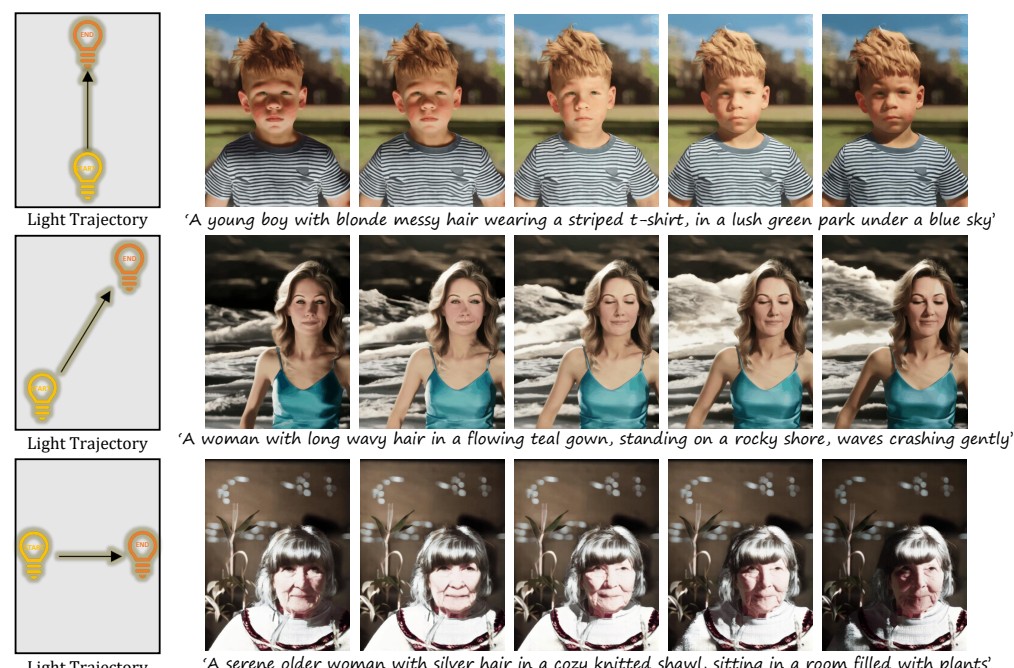

Figure 1: *LumiSculpt* allows user-specified lighting intensity, direction, and trajectories, with textual conditions as input. Being trained once, *LumiSculpt* is capable of generating diverse results at inference time.

## ABSTRACT

Lighting plays a pivotal role in ensuring the naturalness of video generation, significantly influencing the aesthetic quality of the generated content. However, due to the deep coupling between lighting and the temporal features of videos, it remains challenging to disentangle and model independent and coherent lighting attributes, limiting the ability to control lighting in video generation. In this paper, inspired by the established controllable T2I models, we propose *LumiSculpt*, which, for the first time, enables precise and consistent lighting control in T2V generation models.*LumiSculpt* equips the video generation with strong interactive capabilities, allowing the input of custom lighting reference image sequences. Furthermore, the core learnable plug-and-play module of *LumiSculpt* facilitates remarkable control over lighting intensity, position, and trajectory in latent video diffusion models based on the advanced DiT backbone.Additionally, to effectively train *LumiSculpt* and address the issue of insufficient lighting data, we construct *LumiHuman*, a new lightweight and flexible dataset for portrait lighting of images and videos. Experimental results demonstrate that *LumiSculpt* achieves precise and high-quality lighting control in video generation.

# 1 INTRODUCTION

*"If a video tells a story, then lighting is the voice that shapes its tone and mood."*

Lighting is essential for the essential of video generation, which is one of the defining factors for the overall aesthetic quality of the generated video, and is also used to convey emotions, highlight character traits, and guide the audience's attention. Mainstream video generation methods currently employ latent diffusion models (LDMs) to achieve video generation through multi-step denoising in latent space. Research on controllable image and video generation based on LDMs supports our studies of consistent lighting control. Several methods (Ho et al., 2022b; Chen et al., 2023; Ho et al., 2022a; Wang et al., 2023a; Guo et al., 2024) have been developed to achieve relatively accurate text-controlled video generation, as well as video editing (Chai et al., 2023; Ceylan et al., 2023), customization (Wu et al., 2023), and controlling (Wang et al., 2023b; Wei et al., 2024; He et al., 2024). These works have improved the controllability, aesthetics, and usability of video generation. However, due to the deep coupling between lighting and the temporal features of videos, it is challenging to model independent and coherent lighting attributes, resulting in a lack of handy approaches to controlling lighting in videos.

The challenge of customizing lighting lies in three aspects: the lack of training data, the representation of lighting, and the mechanism of injecting lighting features without influencing other attributes. Specifically, although there are currently relighting datasets based on light stages (Debevec et al., 2000), the data format of light stages is not easily applicable in video generation scenarios. Therefore, a flexible dataset that is adaptable to text-controlled new content generation is needed. Obtaining the projection of lighting on the camera's imaging plane requires knowledge of the lighting and the surface texture of the illuminated object (Kim et al., 2024; Ren et al., 2024; Mei et al., 2024), which cannot be satisfied in an end-to-end video generation scenario. Thus, a simpler lighting representation that is only related to lighting parameters is important. Finally, similar to most control tasks, lighting control faces the problem of the deep decoupling of lighting from other elements, such as semantics and color.

In this paper, we propose *LumiHuman* to solve the problem of limited training data. *LumiHuman* is a portrait lighting dataset that can constitut more than 220K videos of humans with known lighting parameters. This is a lightweight and flexible dataset that is not limited to specific lighting movements but is presented in freely combinable frames, laying the foundation for a more diverse range of lighting paths and combinations. We then use virtual engine rendering with known lighting parameters to obtain projections of different directional lighting on planes as a lighting representation. To achieve video lighting control, we propose a consistency lighting control network, *LumiSculpt*, which learns an accurate plug-and-play lighting module capable of controlling the direction and movement of lighting in video generation. To solve the problem of lighting feature injection, we introduce a light control module that takes the lighting projection as input and integrates lighting control injected into the generative model layer by layer. Furthermore, to better decouple lighting and appearance, we design a decoupling loss based on a dual-branch structure, preserving diverse generative capabilities. We implemented *LumiSculpt* on Open-Sora (Lab & etc., 2024) to enable precise lighting control. We conducted comprehensive quantitative and qualitative evaluations. The experimental results show that *LumiSculpt* has achieved state-of-the-art performance in the control of text-to-video lighting, as shown in Figure 1. In summary, our main contributions are as follows:

- We introduce a portrait lighting dataset *LumiHuman*. *LumiHuman* is a continuous lighting video dataset comprising over 220K different videos (*i.e.,* 2.3 million images). *LumiHuman* includes over 30K lighting positions, and over 3K lighting trajectories for each individual. *LumiHuman* paves the way for more lighting control in both image and video generation.

- We introduce *LumiSculpt*, enabling control of lighting direction and movement trajectories in video generation. We propose a lighting representation method, a lighting injection approach, and a lighting decoupling loss for the text-to-video generation scenario, enabling diverse content generation with limited data.

- Extensive experiments prove that *LumiSculpt* has achieved state-of-the-art performance. *LumiSculpt* can not only generate accurate lighting intensity, direction, and trajectories but also maintain coherence and content diversity.

## 2 RELATED WORKS

### 2.1 RELIGHTING

In recent years, deep learning techniques have made significant progress in portrait relighting (Kim et al., 2024; Mei et al., 2023; 2024; Nestmeyer et al., 2020; Pandey et al., 2021; Sun et al., 2019; Wang et al., 2020; Yeh et al., 2022; Zhang et al., 2021), often relying on paired data captured by light stage systems (Debevec et al., 2000) for supervised learning. Typically, these methods require the use of high dynamic range (HDR) environmental maps as input. This process involves estimating intermediate surface properties, including normal vectors, albedo, diffuse reflectance, and specular reflection characteristics. However, the reliance on HDR environmental maps limits the practical application of these techniques in video generation scenarios. Besides, researchers also explore portrait relighting techniques that do not depend on light stage data (Hou et al., 2021; 2022; Wang et al., 2023c).

Recently, diffusion-based models have brought new research directions to the field. Ren et al. (Ren et al., 2024) propose a three-stage lighting-aware diffusion model called Relightful Harmonization, which aims to provide complex lighting coordination for foreground portraits with any background image. Zeng et al. (Zeng et al., 2024) propose a three-stage portrait relighting method using a fine diffusion model called DiLightNet, which calculates radiance cues to re-synthesize and refine the foreground object by combining the rough shape of the foreground object inferred from the preliminary image. Xing et al. (Xing et al., 2024) propose a natural image relighting method called Retinex-Diffusion, which treats the diffusion model as a black-box image renderer and strategically decomposes its energy function to be consistent with the image formation model. However, there is still a lack of methods for lighting control in text-to-video generation. The most related works are lighting control text-to-image generation methods. LightIt (Kocsis et al., 2024) is an image-guided method for image relighting conditioning on shading estimation and normal maps. IC-Light (Zhang et al., 2024) is an image relighting method to generate harmonized background with the user input foreground. Our method is text-guided and requires only text and target lighting conditions to achieve video lighting control.

### 2.2 TEXT-TO-VIDEO SYNTHESIS AND CONTROLLING

Recently, several researches, such as (Ho et al., 2022b; Chen et al., 2023; Ho et al., 2022a; Wang et al., 2023a; Guo et al., 2024), have adopted diffusion models to create highly realistic video content, utilizing text as conditions in guiding the generation process. These studies focus on ensuring consistency between textual descriptions and the final video output. Addressing the issue of difficulty in precisely describing specific visual attributes through text conditions, some studies have attempted to achieve finer video control by fine-tuning models or introducing additional control parameters. Tune-A-Video (Wu et al., 2023) propose a fine-tuning framework that allows users to customize specific videos. VideoComposer (Wang et al., 2023b) use explicit control signals to guide the temporal dynamics of the video. Gong et al. (Gong et al., 2023) introduce TaleCrafter to handle interactions among multiple characters, featuring layout and structural editing capabilities. He et al. (He et al., 2023) propose a retrieval-based deep guidance method that can integrate existing video clips into a coherent narrative video by customizing the appearance of characters. These studies mainly focus on the appearance of visual content or objects. Several methods (Zhao et al., 2023; Zhang et al., 2023b; Wei et al., 2024; He et al., 2024) learn and controll motion through customized diffusion models. These attempts have made pioneering progress in controlling video in specific aspects. However, there is still a lack of effective solutions for precisely controlling the lighting effects in videos.

## 3 LUMIHUMAN

Our goal is to achieve unified control over video lighting, a challenging task with considerable implications. The primary difficulties are threefold: (1) Dataset Scarcity: there is a significant lack of lighting datasets, particularly for videos, with few annotated examples that explicitly showcase lighting changes and where lighting information is well-defined. (2) Complexity of Lighting Attributes: lighting encompasses various factors, including the type of light source, direction of illumination,

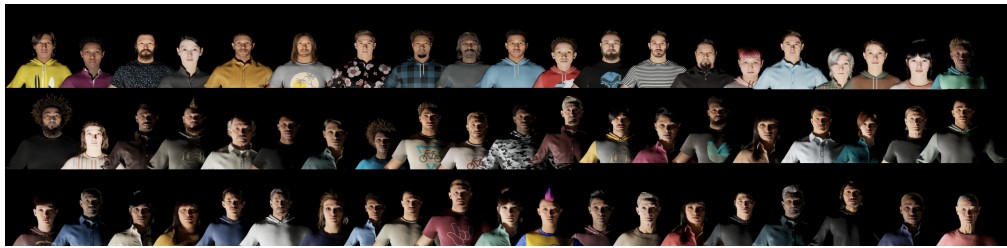

Figure 2: Diverse indivisuals in *LumiHuman*.

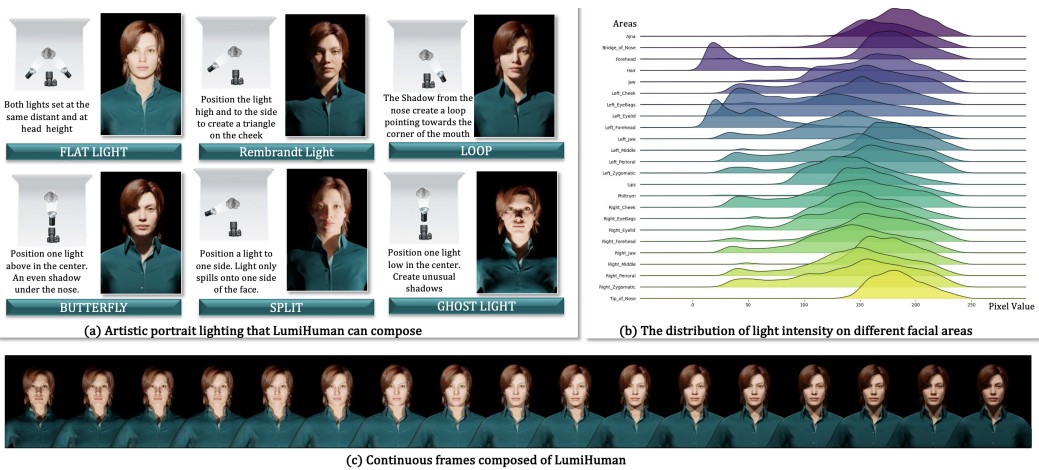

Figure 3: **(a)** *LumiHuman* offers a variety of basic elements that can be combined to form various types of portrait lighting, widely applicable to a range of tasks related to character lighting. **(b)** shows the distribution of light intensity on different facial areas of the characters; *LumiHuman*'s lighting matrix can cover all areas of the face and produce a significant range of light and shadow variations. **(c)** shows an example of creating a continuous lighting video using *LumiHuman*.

and the material properties of objects. In the context of text-to-video generation, the material and shape of the generated objects are often unknown. Thus, effectively representing lighting information to accurately convey its visual effects within the camera's field of view poses a substantial challenge. (3) Attribute Decoupling: like many control tasks, lighting control faces the challenge of decoupling specific attributes. A key technical hurdle is how to isolate lighting information from the appearance of the training data, ensuring that the model does not overfit to the training data's appearance or incorporate irrelevant details.

We introduce a portrait lighting dataset, referred to as *LumiHuman*. *LumiHuman* is a continuous lighting video dataset comprising over **220K** different videos (*i.e.,* **2.3 million** images). The resolution of each video is $1024 \times 1024$. *LumiHuman* is created using Unreal Engine (Epic Games, 2024) for lighting simulation, allowing for the production of data with known lighting information. As shown in Figure 2, *LumiHuman* includes 65 diverse human subjects, 30K lighting positions, and over 3K lighting trajectories for each people.

**Details** As shown in Figure 3(a), *LumiHuman* can be combined to form various types of character lighting. The 30K lighting positions of *LumiHuman* can create light and shadow effects in **all areas** of the human face. As shown in Figure 3(b), we present the brightness distribution map of different regions of the human face. Each ridge in the ridge plot represents a different facial area, with the horizontal axis indicating brightness and the vertical axis indicating the number of samples at the corresponding brightness. *LumiHuman* covers all areas of the face and distributes samples across a wide range of brightness levels. As shown in Figure 3(c), we show the continuous video frames composed of *LumiHuman* samples. *LumiHuman* can form a variety of lighting trajectories flexibly according to user needs, such as horizontal, vertical, diagonal, arc, multi-light source superposition, and so on.

(a) 3D Light Matrix  (b) Generated light moving videos  (c) Annotate the video with text  (d) Caption augmentation

Figure 4: The collection process of *LumiHuman* includes: (a) designing a 3D point light source matrix of $33 \times 33 \times 33$ lighting points, (b) rendering single-frame images and generating portrait lighting videos with various path lighting and lighting reference videos, (c) annotating with a BLIP model, and (d) producing enhanced background captions using a large language model.

**Lighting Representation**   To describe the effect of the lighting projection, a straightforward approach is to incorporate the lighting parameters as additional information in the model. However, this method requires a substantial amount of annotated data to establish a mapping between the lighting vectors and the two-dimensional plane. To better align with the model's inferential feature space, we propose projecting lighting information into an empty space, as illustrated in Figure 5(b). For different lighting positions, this is represented as an image where brighter areas indicate stronger illumination, and darker areas signify weaker lighting. This representation allows for a more effective alignment of lighting information with the video generation model.

**Data Collection**   The *LumiHuman* collection comprises five key stages: **(1) Lighting Design**: As illustrated in Figure 4(a), we developed a lighting position matrix, i.e., a three-dimensional grid measuring $160cm \times 160cm \times 160cm$. The points are uniformly spaced at $5cm$ intervals to serve as lighting positions. Point light sources move within these grid points to capture data of the subject being illuminated from various angles. **(2) Lighting Trajectories Design**: Within the three-dimensional grid, we defined horizontal, vertical, and diagonal trajectories, each composed of grid points to simulate diverse lighting change effects. **(3) Character Construction**: To facilitate the production of portrait lighting data, we utilized the MetaHuman dataset (MetaHuman, 2023), which features 3D models of 65 different individuals. This diversity enhances the visual effects of light projection across different characters. **(4) Flexibility and Storage Optimization**: To achieve a wide variety of lighting path variations while addressing storage concerns, i.e., given the presence of duplicate frames at the same lighting positions, we offer a flexible and lightweight image-video dataset. This dataset includes images rendered from various lighting positions in a three-dimensional grid, where each character corresponds to $33 \times 33 \times 33$ different sampled images. In practice, videos can be generated from images in the dataset using predefined paths, or additional paths can be designed to simulate different lighting effects, as shown in Figure 4 (b). **(5) Text Annotation and Augmentation**: For automatic text annotation of the videos, we employed BLIP (Li et al., 2022). Additionally, we utilized GPT-4 (Achiam et al., 2023) for caption augmentation, generating diverse contextual backgrounds for the dataset, as shown in Figures 4(c) and (d).

## 4  LUMISCULPT

### 4.1  PRELIMINARY

**Text-to-video diffusion models**   Recently, significant advancements have been made in Text-to-Video (T2V) diffusion models (Chen et al., 2023; Wang et al., 2023a; Lab & etc., 2024; Hong et al., 2022). Most of these models follow the classic algorithmic framework from the field of image generation. Specifically, the process typically involves gradually introducing noise $\epsilon$ into $N$ sequences of images $z_1, \ldots, z_N$ until they approximate a Gaussian distribution. Given noise-corrupted inputs $z_1, \ldots, z_N$, a neural network is trained to predict the added noise. During training, the network strives to minimize the mean squared error (MSE) between its noise predictions and the actual noise.

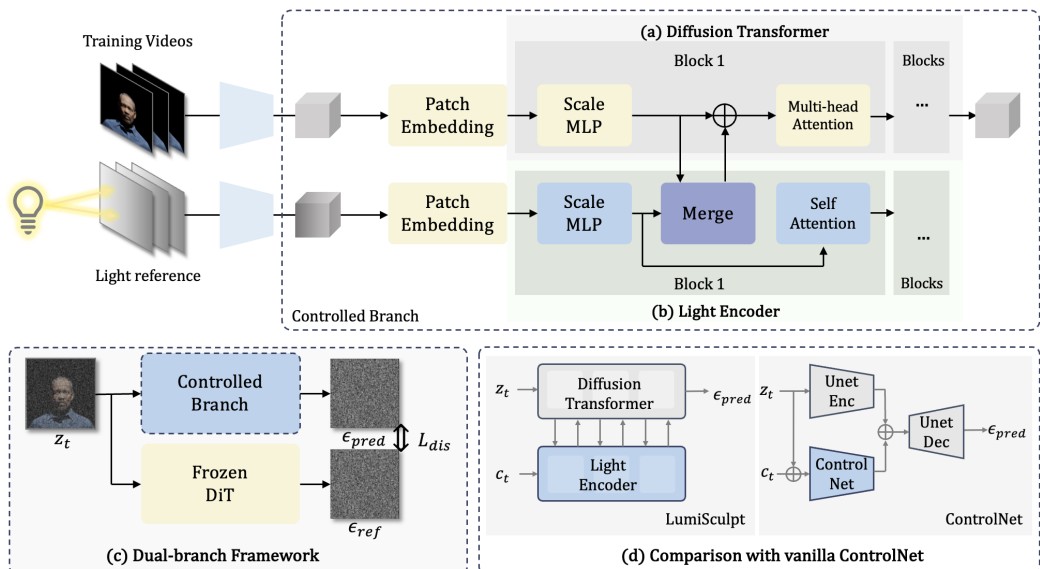

Figure 5: The pipeline of *LumiSculpt* consists of the generation backbone, i.e. the controlled branch, which includes (a) a diffusion transformer (DiT), a pre-trained video denoising network, and (b) a light encoder, a trainable external transformer network. The light encoder takes light reference latents as input and processes them through various blocks to produce a light condition sequence. This sequence is integrated into the generation backbone using several merge modules within each block. During training, we propose a **(c) dual-branch framework** including a controlled branch and a frozen branch, which provide regularization for diverse appearances. The frozen branch is a DiT with frozen parameters, sharing weights with (a). Both branches predict noise, resulting in $\epsilon_{pred}$ and $\epsilon_{reg}$, which are used to compute the disentanglement loss $\mathcal{L}_{dis}$. (c) and (d) show that *LumiSculpt* differs form ControlNet (Zhang et al., 2023a) in terms of model structure, condition injection, training manners and objectives.

## 4.2 Integrating Lighting into Video Generators

Since lighting is represented in pixel space, it can be parameterized as input to the standard visual model. We extract lighting features using a Variational Autoencoder (VAE) shared with a generative model, which are then fed into a dedicated lighting encoder. As illustrated in Figure 5(b), this lighting encoder employs a transformer architecture comprised of self-attention layers, enabling it to compute attention scores globally across the video. This design allows the encoder to effectively capture the spatial and temporal relationships of lighting throughout the video clip. The lighting encoder accepts lighting features as input. The transformer blocks, matching the number of layers in the backbone model, output a sequence of latents of the same size. For each layer in the backbone model, the lighting encoder provides corresponding features of identical dimensions, facilitating feature fusion. Our objective is to seamlessly integrate these latents into the DiT architecture of the T2V model. The latent features of the video, denoted as $z_t$, and the lighting features, $c_t$, are combined through element-wise addition. This integrated feature is then passed through a linear layer, producing the output for the subsequent layer with a hyper-parameter termed *guidance scale* set to 0.5. As shown in Figure 5(d), *LumiSculpt* employs 3D self-attention mechanisms as the lighting encoder and uses multi-stage weighting as condition injection mechanisms. ControlNet uses the U-Net Encoder to extract features and injects conditions by adding latents.

## 4.3 Lighting Learning

We utilize a data-driven approach to learn complex lighting information, capturing the projection effects of lighting at different positions on the human face. The implementation of this method is based on the *LumiHuman*. There is an issue with simply learning lighting from data rendered by unreal engines, which is the leakage of appearance information. This occurs due to the consistent

backgrounds and layouts in the dataset. While such consistency provides more stable training data for lighting learning, it simultaneously increases the model's susceptibility to overfitting to similar appearances. This raises the key issue of lighting control: how to decouple lighting from other elements? We propose a lighting-appearance disentanglement method, which includes a dual-branch framework and a novel disentanglement loss.

**Dual-Branch Framework**   To obtain diverse appearance data, a naive solution is to use additional video data as regular samples to provide the model with diverse appearances. However, it is challenging to obtain a large amount of diverse data with known lighting conditions, so we opt for the manufacture of regular samples based on generative models. As shown in Figure 5(c), we propose a dual-branch structure, including a training branch and a frozen branch, by introducing a frozen foundational denoising model to provide an appearance reference. During training, the dual branches accept the same textual conditions and noisy latents, obtaining predicted noises $\epsilon_t$ and $\epsilon_t^{reg}$, respectively. The diverse appearances of the pre-trained model are reflected in $\epsilon_t^{reg}$. In this way, we achieve low-cost regular sample manufacturing.

**Loss Functions**   We train the model to learn the overall distribution of the dataset using a simple denoising loss $\mathcal{L}_{denoise}$, which includes lighting information and appearance information from the training videos, the latter of which we do not require. To remove appearance, a quantitative method (e.g., a loss function) capable of measuring appearance consistency without affecting lighting learning is needed. On the one hand, this loss should be able to quantify the appearance consistency of the latents from two videos, and on the other hand, the loss must be independent of planar location, otherwise it will affect the learning of lighting distribution. Inspired by style transfer (Jing et al., 2020), which also needs to measure style consistency without being disturbed by structured information, we propose a disentanglement loss $\mathcal{L}_{dis}$ to measure appearance consistency. We use AdaIN (Huang & Belongie, 2017) to measure the feature distribution patterns of two videos' latents, reflecting their appearance feature distribution. Aligning the feature distribution of the training branch-generated video with the frozen branch retains the diverse appearance of the original model. With this approach, the generative model globally aligns with the training video under the drive of $\mathcal{L}_{denoise}$, while squeezing out redundant appearance information under the drive of $\mathcal{L}_{dis}$, retaining the generative capability of the original model. The training objective function can be represented as:

$$\mathcal{L}_{dis} = ||(\sigma(z_0^{pred}) - \sigma(z_0^{reg})||_2 + ||(\mu(z_0^{pred}) - \mu(z_0^{reg})||_2,$$
$$\mathcal{L}_{denoise} = \mathbb{E}_{z_{1:N},\epsilon,c_t,t}\left[||\hat{\epsilon}(z_{1:N}^{pred},c_t,t) - \epsilon||^2\right], \quad (1)$$
$$\mathcal{L}_{total} = \mathcal{L}_{denoise} + \beta\mathcal{L}_{dis},$$

where $\sigma(\cdot)$ and $\mu(\cdot)$ denote the computation of standard variance and mean, respectively. $z_0^{reg} = z_t^{reg} - \epsilon_{reg}$ represents the predicted denoised output of the frozen branch at time step $t$, while $z_0^{pred} = z_t^{pred} - \epsilon_{pred}$ signifies the predicted denoised output of the controlled branch at time step $t$. $N$ denotes the total number of steps, and $c_t$ represents the textual condition. $\beta$ is set to 3.0.

## 5 EXPERIMENTS

### 5.1 EXPERIMENTAL SETUP

**Methods for comparison**   We compare our approach with state-of-the-art text-to-video generation methods Open-Sora (Lab & etc., 2024), image relighting method IC-light (Zhang et al., 2024), and image control method ControlNet (Zhang et al., 2023a).

**Metrics**   We employ a variety of quantitative and qualitative metrics to assess the lighting accuracy, inter-frame coherence, and visual-text similarity of generated videos.

**Evaluation dataset**   We use 500 different light paths and captions not present in the training dataset as conditions to guide the comparative methods in generating evaluation videos.

**Implementation details.**   In all video generation experiments, we use Open-Sora v1.2.0 (Lab & etc., 2024) with the default network architecture. We set a learning rate of $1 \times 10^{-4}$. The input

video resolution is $640 \times 480 \times 29$. The training process for each motion requires approximately $800 \sim 1500$ iterations using eight NVIDIA A100. The number of inference steps is set to $T = 50$ and the guidance scale is set to $w = 7.5$.

Table 1: Quantitative experimental results and ablation study results.The best results are marked as **bold** and the seconds one are marked by underline.

| Method | Consistency | | Lighting Accuracy | | Quality |
|---|---|---|---|---|---|
| | CLIP↑ | LPIPS↓ | Direction↓ | Brightness↑ | CLIP↑ |
| Open-Sora | 0.9845 | 1.3503 | 0.4542 | 0.8229 | 0.3182 |
| IC-Light | 0.9703 | 2.5329 | 0.5264 | 0.8632 | 0.3145 |
| ControlNet | 0.8081 | 5.9324 | 0.5500 | 0.8032 | 0.3440 |
| Ours(full model) | 0.9951 | 1.1312 | 0.3500 | 0.8779 | **0.3597** |
| Ours(w/o caption aug) | 0.9948 | 1.1211 | 0.2992 | **0.9269** | 0.3416 |
| Ours(w/o $\mathcal{L}_{dis}$) | **0.9957** | **1.1033** | **0.1945** | 0.8363 | 0.2909 |

## 5.2 QUANTITATIVE EVALUATIONS

As shown in Table 1, we use five quantitative metrics for evaluation: (1) Frame-wise CLIP image similarity: we utilize the similarity of frame-wise CLIP (Radford et al., 2021) image embeddings to evaluate the semantic-level video coherence. A higher value indicates greater inter-frame similarity, suggesting better semantic stability in the generated video. (2) Frame-wise Learned Perceptual Image Patch Similarity (LPIPS) consistency: we measure feature-level coherence using frame-wise LPIPS consistency. A lower value signifies smaller feature discrepancies, indicating higher inter-frame consistency. (3) Light directions Root Mean Squared Error (RMSE): we calculate the lighting direction for each frame and then assess the consistency of the generated video's lighting direction with the reference. RMSE represents the discrepancy in lighting direction; the smaller the value, the more consistent the lighting with the target. (4) Brightness consistency: we segment each video frame into patches, compute the average brightness of different patches, and construct a brightness distribution relationship. This distribution is only related to the comparison between different patches and is independent of the absolute brightness values. We calculate the brightness distribution consistency between the generated video and the reference video. (5) CLIP text-image similarity: we measure the model generation quality using the similarity between the clip image embedding of the generated video frames and the text embedding of the caption. The higher the similarity, the better the generation quality. It can be observed that, compared to Open-Sora and IC-Light, *LumiSculpt* is capable of maintaining good inter-frame consistency and text-image consistency while achieving accurate lighting control.

## 5.3 QUALITATIVE EVALUATIONS

As shown in Figure 6, due to the absence of video illumination control methods, we compare our approach with the image illumination control method IC-Light based on diffusion models, and the video generation method Open-Sora. We consider two light intensity levels, strong and soft, as well as horizontal and vertical lighting movement directions. Since IC-Light is designed for relighting existing images, we use portraits generated by our method as foreground guidance. IC-Light is capable of producing single-frame images with accurate lighting directions, but due to a lack of inter-frame awareness, the coherence of the output video is poor, with noticeable flickering in the background. Open-Sora can generate coherent and aesthetically pleasing videos, but struggles to control lighting direction via textual conditions, resulting in relatively unchanged lighting throughout the video. Our method not only ensures video coherence and visual quality but also achieves precise control over lighting trajectory and intensity. Video results are provided in the supplementary material.

## 5.4 ABLATION STUDY

As shown in the $1^{st} \sim 3^{rd}$ and last rows of Figure 7, we present the results of ablating different modules of *LumiSculpt*. Removing caption augmentation from *LumiHuman* leads to a lack of diverse textual guidance, causing the model during training to rely solely on text conditions that exactly match the dataset, thus improving appearance fitting. As shown in the second row, the generated results exhibit consistent pose and layout. Without the dual-branch structure and decoupling loss,

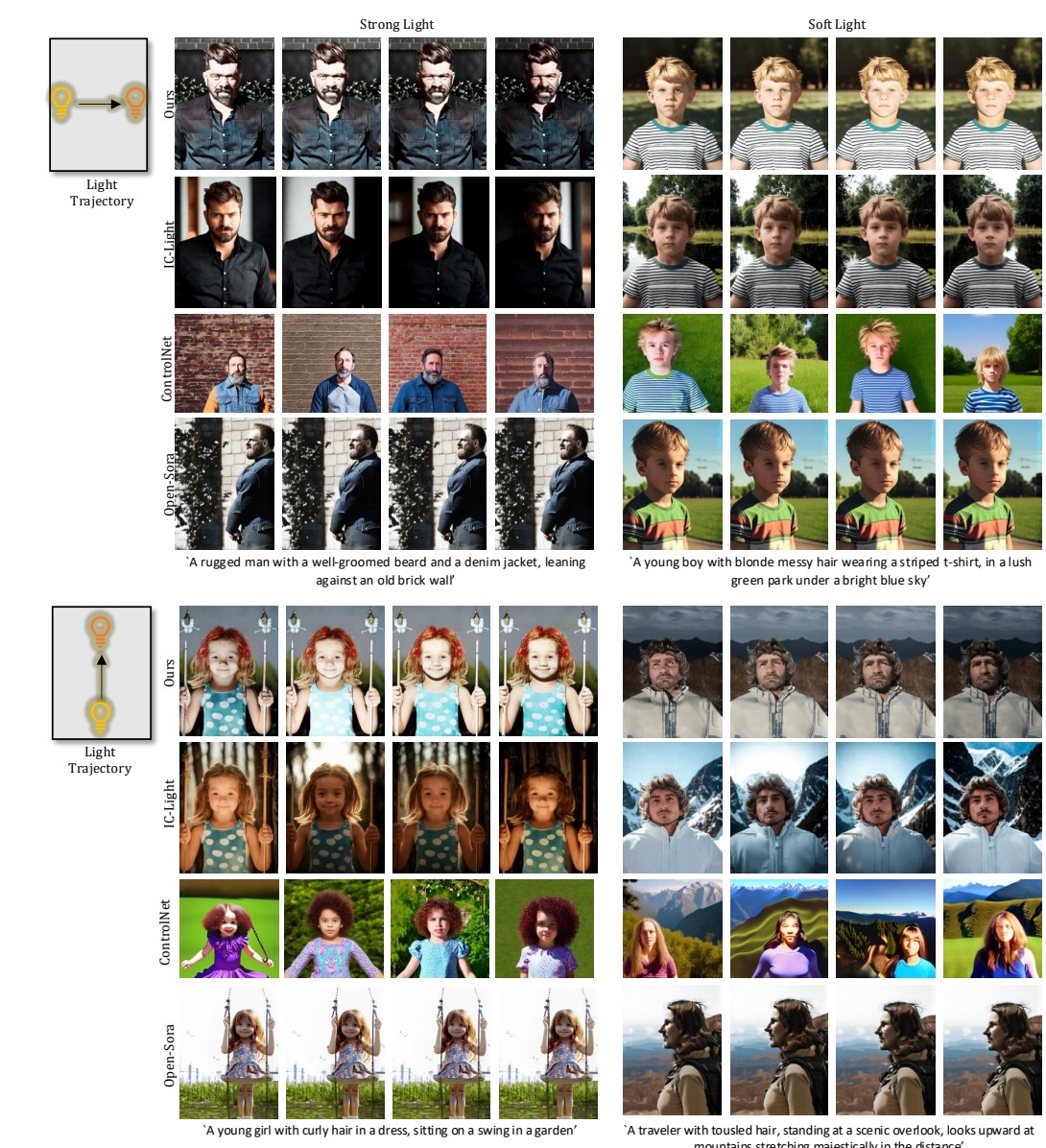

Figure 6: Comparison results with state-of-the-art methods IC-Light (Zhang et al., 2024), Control-Net (Zhang et al., 2023a) and Open-Sora (Lab & etc., 2024). The classic horizontal and vertical directions for light movement and two brightness levels are tested to achieve a comprehensive qualitative evaluation.

Table 2: Experimental results of hyper-parameter *guidance scale*. The best results are marked as **bold** and the seconds one are marked by underline

| Scale | Consistency↑ | Accuracy↓ | Quality↑ |
|---|---|---|---|
| *scale*=0.1 | 0.9943 | 0.4239 | 0.3163 |
| *scale*=0.3 | **0.9964** | 0.3825 | **0.3814** |
| *scale*=0.5 | 0.9951 | 0.3500 | 0.3597 |
| *scale*=0.7 | 0.9939 | **0.2484** | 0.3499 |
| *scale*=0.9 | 0.9902 | 0.2922 | 0.2941 |

as shown in the third row, the generated appearances tend to overfit the training data, making it challenging to produce diverse backgrounds. As illustrated in the first row, the complete *LumiSculpt* successfully balances diverse appearances with accurate lighting. As shown in the $4^{th} \sim 7^{th}$ rows of Figure 7 and Table 2, we present both qualitative and quantitative analysis results of varying the

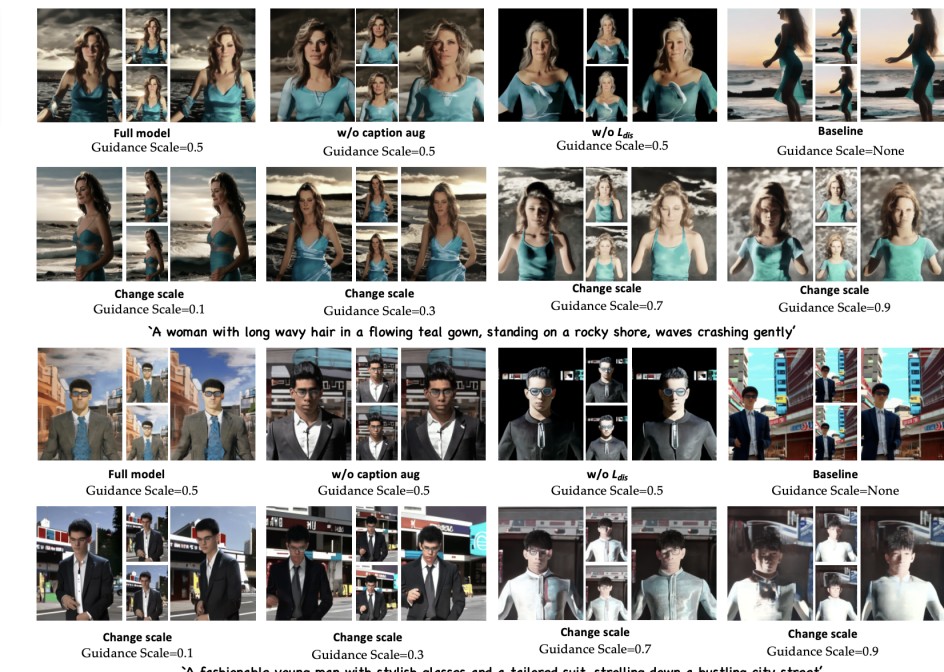

Figure 7: Ablation study results. We conducted ablations on key components related to model training, including caption augmentation and $\mathcal{L}_{dis}$, both of which contribute to the diversity of generated content. We presented results with varying values of the important parameter *guidance scale*, which affects the accuracy of lighting and the diversity of appearance, related to model inference.

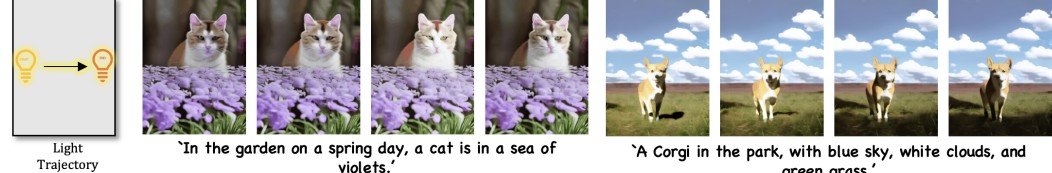

Figure 8: The results of *LumiSculpt* on animals. *LumiSculpt* is capable of learning lighting priors and transferring them to non-human objects.

hyper-parameter *guidance scale*. During inference, the standard *LumiSculpt* sets the guidance scale to 0.5. Increasing the guidance scale intensifies the strength of lighting guidance, enhancing the accuracy of lighting direction and brightness, but an excessively large guidance scale can undermine the model's ability to generate diverse outputs. In contrast, decreasing the guidance scale reduces the strength of lighting guidance, leading to a decrease in lighting accuracy.

**Generalization**    As shown in the Figure 8, *LumiSculpt* provides lighting priors on animals, which demonstrates the generalization ability to real-world cases.

## 6 CONCLUSION

In this paper, we address the challenge of lighting control in text-to-video generation. To address data scarcity, lighting representation difficulties, and lighting injection complexities, we introduce a flexible portrait lighting dataset, *LumiHuman*, along with a plug-and-play lighting guidance method, *LumiSculpt*. Our proposed dual-branch structure and associated loss function for decoupling are not only effective for lighting control but also have the potential to be generalized across a variety of generative tasks. As video generation techniques continue to evolve, enabling precise lighting control becomes an increasingly important research direction, driven by significant aesthetic demands from both professionals and the general public. We believe that *LumiHuman* and *LumiSculpt* will serve as valuable resources and methodologies for future explorations in this domain.

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

# Rebuttal for "*LumiSculpt*"

ICLR 2025,

Manuscript ID: 4419

## [Common Concern 1] Statistics and Diversity of *LumiHuman* Dataset

We introduce the *LumiHuman* dataset, a continuous lighting video dataset comprising over **220K** different videos (*i.e.,* **2.3 million** images). The resolution of each video is $1024 \times 1024$. As shown in Fig. 1, *LumiHuman* includes 65 diverse human subjects, 30K lighting positions, and over 3K lighting trajectories for each people.

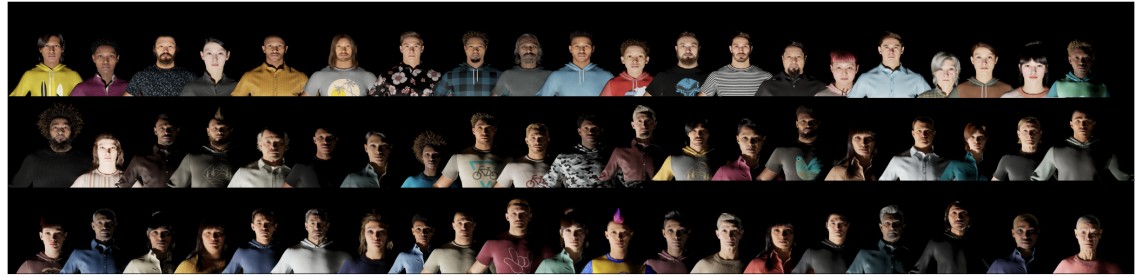

(a) Divers individuals in LumiHuman

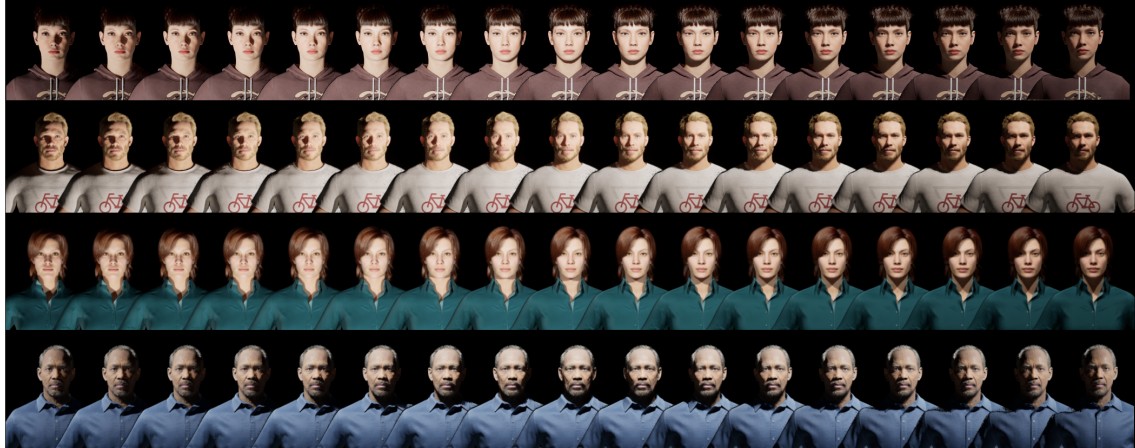

(b) Frames with various lighting trajectories in LumiHuman

**Fig. 1.** Samples in *LumiHuman*.

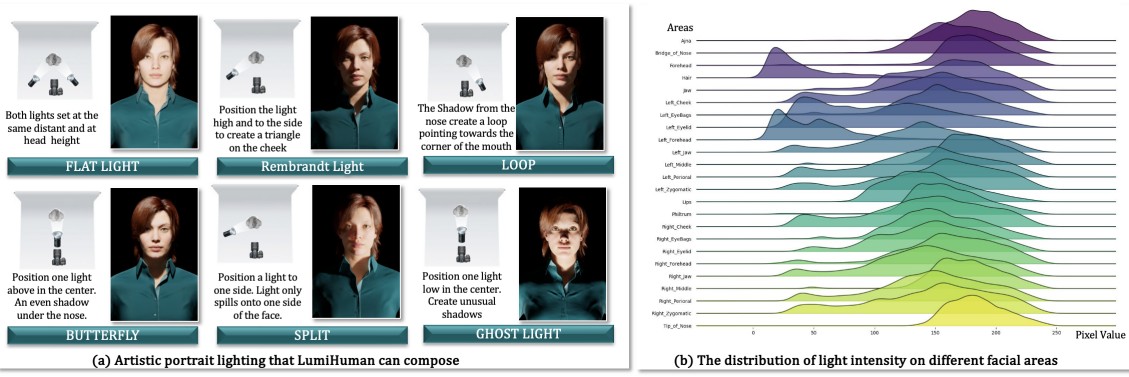

(a) Artistic portrait lighting that LumiHuman can compose

(b) The distribution of light intensity on different facial areas

**Fig. 2.** Illustration of the light sources and camera, and the ridge plot of illuminated areas (*i.e.*, face patches.)

Table I: Comparison of other lighting-related datasets.

| Dataset | Synthesis | Light Positions | Light Movement | Number of Images | Subject | Resolutions |
|---|---|---|---|---|---|---|
| DPR | 2D | 7 | None | 138K | - | $1024 \times 1024$ |
| Openillumination | Light Stage | 142 | None | 108K | 64 objects | $3000 \times 4096$ |
| LumiHuman | 3D | **35,937** | **>3K** | **2.3M** | **65 indivisuals** | $1024 \times 1024$ |

Our *LumiHuman* of 65 human identities is sufficient for training *LumiSculpt*, which is supported by extensive qualitative and quantitative experiments. The scalability of synthetic data lies in the ability to construct diverse light trajectories, leveraging varied lighting data to facilitate the model's learning of illumination harmonization.

**[Common Concern 2] Similar to ControlNet**

*LumiSculpt* is distinct from ControlNet in terms of its task, motivation, module design, training objective, training data, backbone, and generated results. A detailed explanation for each point is provided below:

- **Task:** *LumiSculpt* is a specialized lighting control method designed for DiT based T2V models. ControlNet is a control method that focuses on image geometry (pose, depth map, canny, etc.) for U-Net based T2I models.
- **Motivation:** *LumiSculpt*'s motivation focuses on elements in videos that affect realism and aesthetics, i.e., lighting, and proposes a method to achieve coherent video generation with controllable lighting. ControlNet's motivation stems from the randomness in T2I diffusion models, hence it introduces a method for generating images with controllable geometry.
- **Module Design:** As shown in Fig. 3(d), *LumiSculpt* employs self-attention mechanisms as the lighting encoder and uses linear layers and latent weighting as condition injection mechanisms. ControlNet uses the U-Net Encoder to extract features and injects conditions by adding latents. These atomic components are commonly used and necessary for feature extraction and condition injection, which are not limited to a specific method.
- **Training Objective:** *LumiSculpt* tackles the core challenge of the entanglement of lighting and appearance. As shown in Fig. 3(c), *LumiSculpt* employs a dual-branch structure and an appearance-lighting disentanglement loss. ControlNet is trained with the diffusion noise prediction loss.
- **Training Data:** *LumiSculpt* utilizes video data with coherent inter-frame lighting changes, whereas ControlNet is based on independent images.
- **Backbone:** *LumiSculpt* is build upon DiT-based Open-Sora-Plan (Lab & etc., 2024), and ControlNet is designed for U-Net structured Stable Diffusion (Rombach et al., 2022).
- **Generated Results:** *LumiSculpt* generates coherent videos while ControlNet generates images.

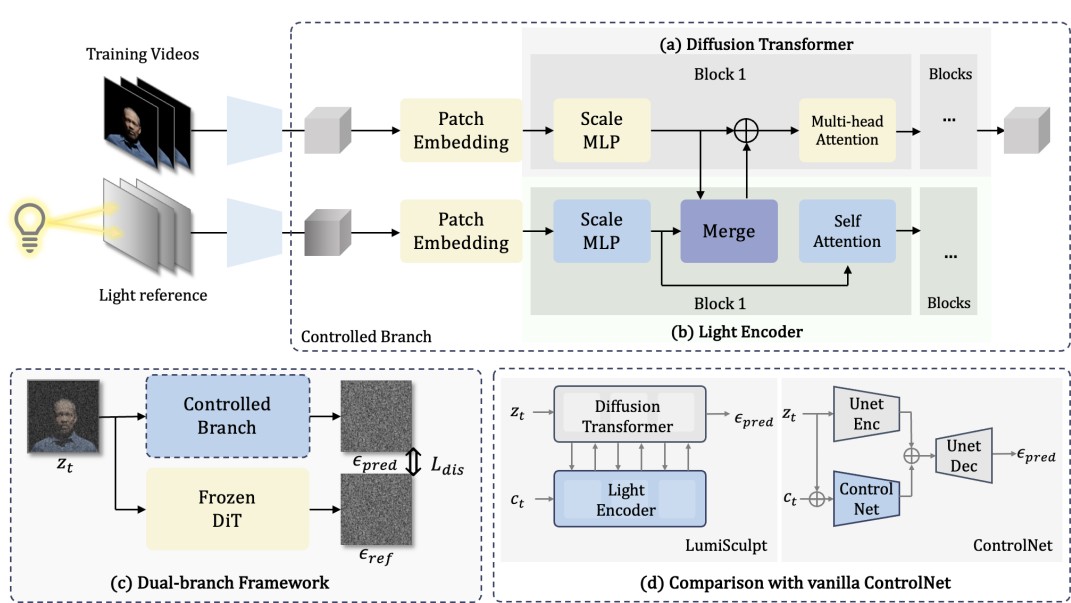

**Fig. 3.** Differences between *LumiSculpt* and ControlNet.

We implement ControlNet to video lighting control by training with paired frames in *LumiHuman* and generating image sequence as video. The comparison results are shown in Fig. 4 and Tab. II. ControlNet struggles to achieve lighting control, generating images with random lighting. This validates the effectiveness of our model structure and training methodology.

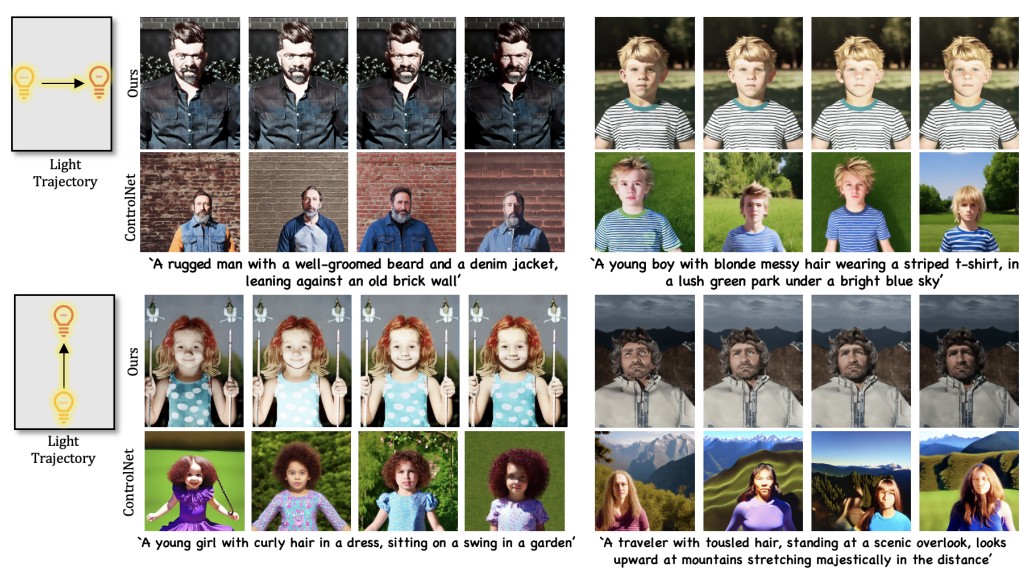

**Fig. 4.** Comparison results with state-of-the-art methods ControlNet (Zhang et al., 2023).

Table II: Quantitative experimental results and ablation study results. The best results are marked as **bold**.

| Method | Consistency | | Lighting Accuracy | | Quality |
|--------|:---:|:---:|:---:|:---:|:---:|
| | **CLIP↑** | **LPIPS↓** | **Direction↓** | **Brightness↑** | **CLIP↑** |
| Open-Sora | 0.9845 | 1.3503 | 0.4542 | 0.8229 | 0.3182 |
| IC-Light | 0.9703 | 2.5329 | 0.5264 | 0.8632 | 0.3145 |
| ControlNet | 0.8081 | 5.9324 | 0.5500 | 0.8032 | 0.3440 |
| Ours | **0.9951** | **1.1312** | **0.3500** | **0.8779** | **0.3597** |

# Referee: #1 rLXS

**Comment #1**

LumiHuman is synthetic, which may limit the model's performance in real-world cases. I wonder if there can be a thorough evaluation of real-world cases. There are only 65 individuals in the dataset, which may limit the model to generalize to new portraits.

**Response:** Thanks for your suggestion. As shown in the Fig. 5, *LumiSculpt* supports the generation of videos featuring diverse backgrounds, environments, and characters and also provides lighting priors on **non-human objects**. This demonstrates the generalization ability to real-world cases.

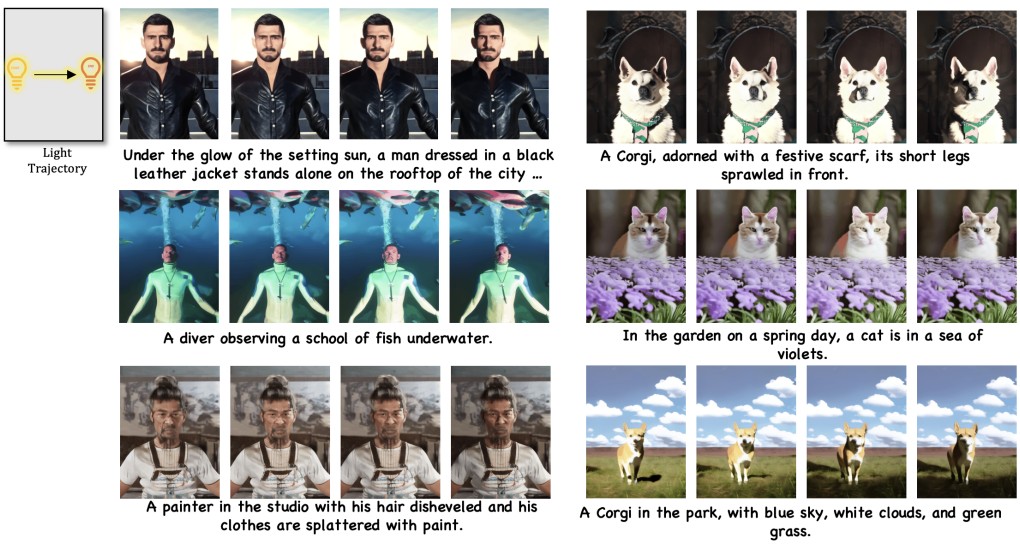

**Fig. 5.** More results with *LumiSculpt*.

Synthetic data does not compromise the model's generalization. During training, *LumiSculpt* employ various strategies to mitigate overfitting, ensuring that the light control module primarily learns the patterns of light variation rather than the appearance of the characters. To evaluate with real-world case, we employ the commonly used FID (Seitzer, 2020) score to assess the photo-realism of both *LumiSculpt* and Open-Sora (Lab & etc., 2024) within the FFHQ (Karras et al., 2019) dataset. As shown in Table III, *LumiSculpt* achieves a better FID score, demonstrating its ability to generate realistic videos.

Table III: FID of *LumiSculpt* and Open-Sora using the FFHQ (Karras et al., 2019) dataset

| Method | Open-Sora | LumiSculpt |
|--------|-----------|------------|
| FID ↓ | 35.7 | **33.0** |

The "*65 individuals*" is also not the limiting factor for model training. *LumiSculpt* learns lighting variation patterns and achieves generalization through diverse light trajectories constructed from synthetic data, rather than relying on human appearances.

**Comment #2**

The generated videos are not informative enough. The motion dynamics are not enough. I wonder if there are results where the portrait and background can move more vividly

**Response:** Thanks for the comment. Yes, with motion descriptions, *LumiSculpt* exhibits motion dynamics where the portrait and background can move more vividly. As shown in Fig. 6, we have marked the regions with significant motion changes. Actually, generating portrait and background with vivid dynamics is challenging for T2V models, and it is even harder to control both lighting and motion dynamics. As illustrated in Fig. 7, applying image-based lighting control methods (since there is no suitable video-based model available) cannot achieve inter-frame consistency. Therefore, *LumiSculpt* provides a novel solution for controllable video generation, particularly focused on lighting.

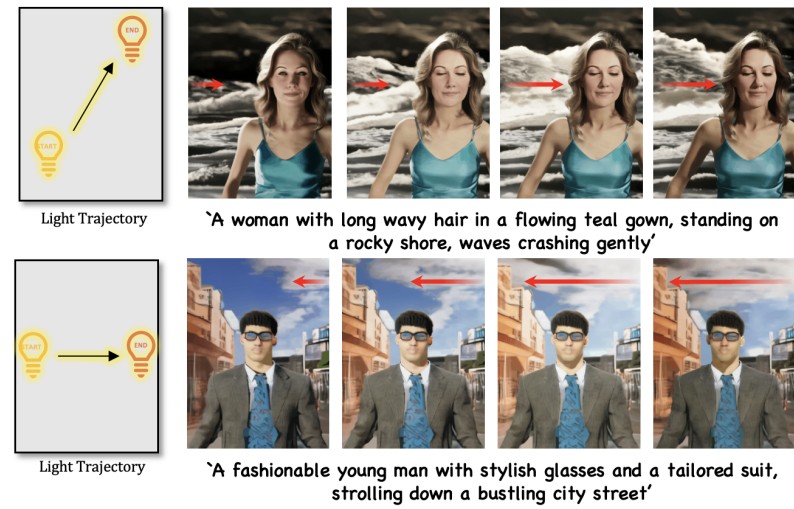

**Fig. 6.** Dynamic videos generated by *LumiSculpt*.

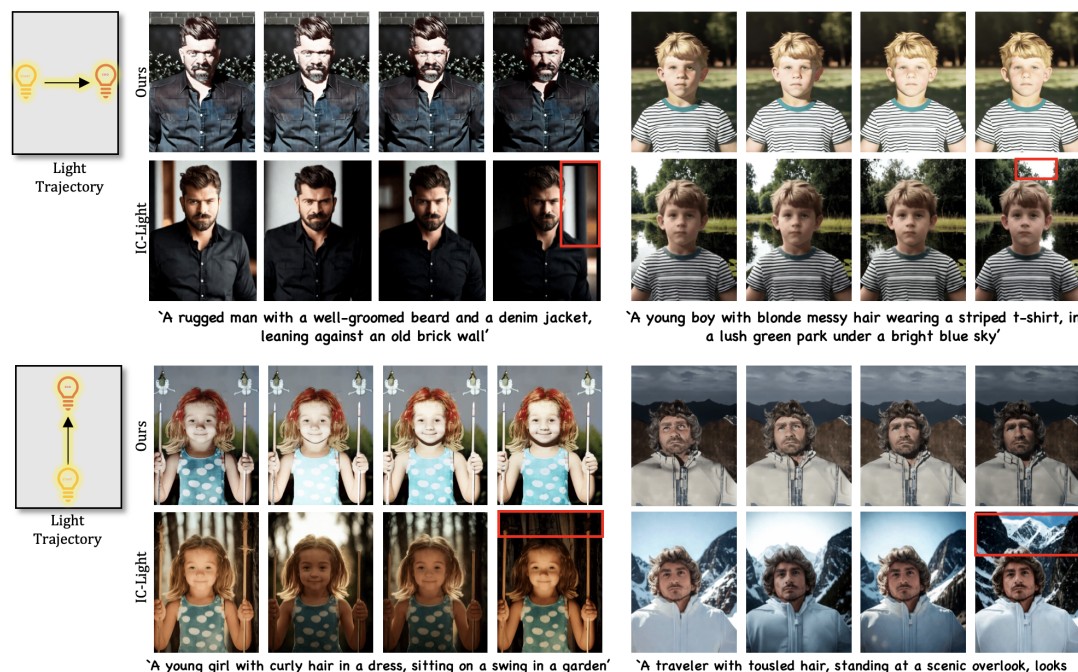

**Fig. 7.** Comparison results with state-of-the-art methods IC-Light (Zhang et al., 2024).

# Referee: #2 MY7D

## Comment #1

This algorithm seems more suitable for image generation, as I did not observe any specific design tailored for video tasks. Video generation is merely an extension of the algorithm's application.

**Response:** Thanks for the comment. **Firstly**, *LumiSculpt* incorporated 3D attention specifically designed for temporal modeling in videos. All light injection modules in this work are built upon the backbone of the video diffusion generation model, ensuring consistent temporal modeling of light dynamics without compromising the model's original generative capabilities. **Secondly**, lighting control in image generation primarily focuses on harmonizing lighting between the background and the subject. When directly applied the image based method to video generation, it may result in severe temporal inconsistencies, as each frame may exhibit different visual content. In contrast, our approach demonstrates smooth and stable lighting across video frames, reflecting the effectiveness of our current design, which including conditional extraction and injection methods, for video generation.

## Comment #2

In the comparisons, the authors use images generated by the network as the foreground. Does this imply that, limited by the synthetic data used during training, the algorithm may not generalize well to real-world scenes? I also noticed unnatural foreground (human) generation results in the video demo.

**Response:** Thanks. Synthetic data does not compromise the model's generalization. During training, *LumiSculpt* also employ various strategies to mitigate overfitting, ensuring that our light control module primarily learns the patterns of light variation rather than the appearance or content of the characters. We employ the commonly used FID (Seitzer, 2020) score to assess the realism of the generated results for both *LumiSculpt* and Open-Sora (Lab & etc., 2024) within the FFHQ (Karras et al., 2019) dataset. As shown in Table IV, *LumiSculpt* achieves a better FID score, demonstrating its ability to generate realistic videos.

Table IV: FID of *LumiSculpt* and Open-Sora using the FFHQ (Karras et al., 2019) dataset

| Method | Open-Sora | LumiSculpt |
|--------|-----------|------------|
| FID ↓ | 35.7 | **33.0** |

*LumiSculpt* is a T2V method, aiming at generating lighting controllable videos by texts. Thus, the ability to generate both foreground and background with text is an advantage of *LumiSculpt*. IC-Light's goal is re-lighting, which involves harmonizing lighting between foreground and background images. Thus, IC-Light's foreground is generated by *LumiSculpt* because it needs a foreground image.

## Comment #3

Can this dataset be open-sourced to ensure reproducibility for future work?

**Response:** Yes, it certainly will be open-sourced upon acceptance.

**Comment #4**

I find the caption augmentation section somewhat unclear. Is it simply replacing captions, or does it involve corresponding changes in the image background as well?

**Response:** It is replacing captions. During training, the augmented captions serve as textual conditions input into the dual-branch models. These captions can guide the frozen branch to produce latents for the same character against different backgrounds, which act as regularization samples providing strong appearance constraints for the $\mathcal{L}_{dis}$. This drives the Controlled Branch to generate richer backgrounds instead of only black backgrounds. As shown in the first and second rows of Fig. 8, the inclusion of augmented captions enhances the model's ability to generate diverse backgrounds and layouts.

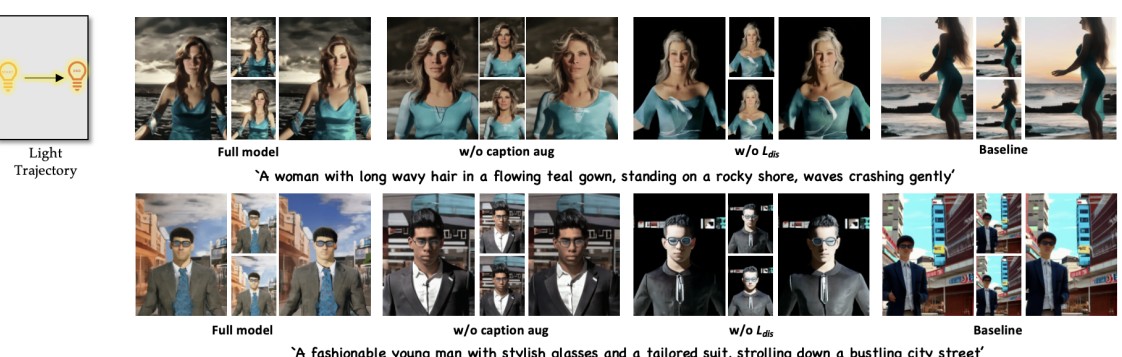

Light Trajectory     Full model     w/o caption aug     w/o $L_{dis}$     Baseline

`A woman with long wavy hair in a flowing teal gown, standing on a rocky shore, waves crashing gently`

Full model     w/o caption aug     w/o $L_{dis}$     Baseline

`A fashionable young man with stylish glasses and a tailored suit, strolling down a bustling city street`

**Fig. 8.** Ablation results of augmented captions.

# Referee: #3 wtpx

**Comment #1**

The synthetic renderings could follow the usual light stage setup with full coverage, not just frontal lighting.

**Response:** We sincerely appreciate your valuable suggestions regarding lighting settings. *LumiHuman* only include light sources in front of the characters, because in an environment with point light sources, the light behind the characters would be blocked by the human body, resulting in a black image, or it appears as a near-white light spot, making it difficult to see the object. These phenomena exist in both generated data and real-world light-stage data (Liu et al., 2024).

Our current light matrix is capable of creating rich light and shadow effects. *LumiHuman* provides over 30K lighting positions and over 3K lighting trajectories for each individual. These lighting positions can create light and shadow effects in **all areas** of the human face. As shown in Fig. 9, we present the brightness distribution map of different regions of the human face. Each ridge in the ridge plot represents a different facial area, with the horizontal axis indicating brightness and the vertical axis indicating the number of samples at the corresponding brightness. *LumiHuman* covers all areas of the face and distributes samples across a wide range of brightness levels.

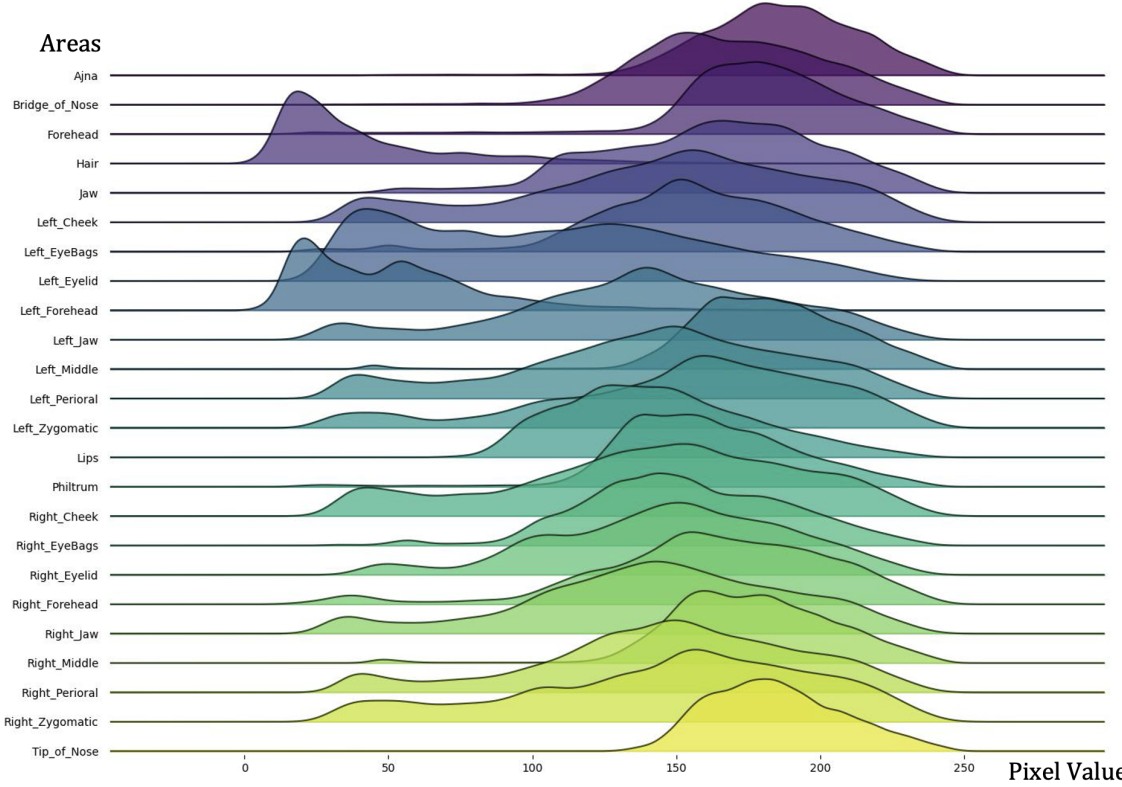

**Fig. 9.** The distribution of light intensity on different facial areas of the characters. *LumiHuman*'s lighting matrix can cover all areas of the face and produce a significant range of light and shadow variations.

**Comment #2**

Furthermore, it is not clear whether 65 identities can provide enough diversity. I believe that the main advantage of using a synthetic dataset is that it can be scaled.

**Response:** Thanks for the comment. Our *LumiHuman* of 65 human identities can provide sufficient diversity to train *LumiSculpt*, which is supported by extensive qualitative and quantitative experiments. The scalability of synthetic data lies in the ability to construct diverse light trajectories, leveraging varied lighting data to facilitate the model's learning of illumination harmonization. As shown in Tab. V, compared to other lighting datasets Openillumination (Liu et al., 2024) and Deep Portrait Relighting (DPR) dataset (Zhou et al., 2019)(generated from face image dataset Celeb-A (Liu et al., 2015)), *LumiHuman* outperforms in light positions, light movements and number of images.

Table V: Comparison of other lighting-related datasets.

| Dataset | Synthesis | Light Positions | Light Movement | Number of Images | Subject | Resolutions |
|---|---|---|---|---|---|---|
| DPR | 2D | 7 | None | 138K | - | 1024 × 1024 |
| Openillumination | Light Stage | 142 | None | 108K | 64 objects | 3000 × 4096 |
| LumiHuman | 3D | **35,937** | **>3K** | **2.3M** | **65 indivisuals** | 1024 × 1024 |

**Comment #3**

It would also be crucial to show that available real-world light-stage datasets cannot provide enough supervision to achieve such quality for lighting control.

**Response:** Thanks for the valuable suggestion. **Firstly**, available public light-stage datasets, e.g., Openillumination (Liu et al., 2024), do not contain human subject data, and its One-Light-At-a-Time (OLAT) data comprises only 142 lighting positions, which is hard to achieve smooth changes in lighting. Relying solely on publicly light-stage datasets is insufficient for T2V model training. **Secondly**, real-world light-stage datasets rely on HDR maps that have a significant domain gap with T2I and T2V scenarios. **In summary**, as shown in Figure 10, *LumiHuman* provides a coordinated, large spatial range of light sources, enabling users to freely combine the types of lighting they require.

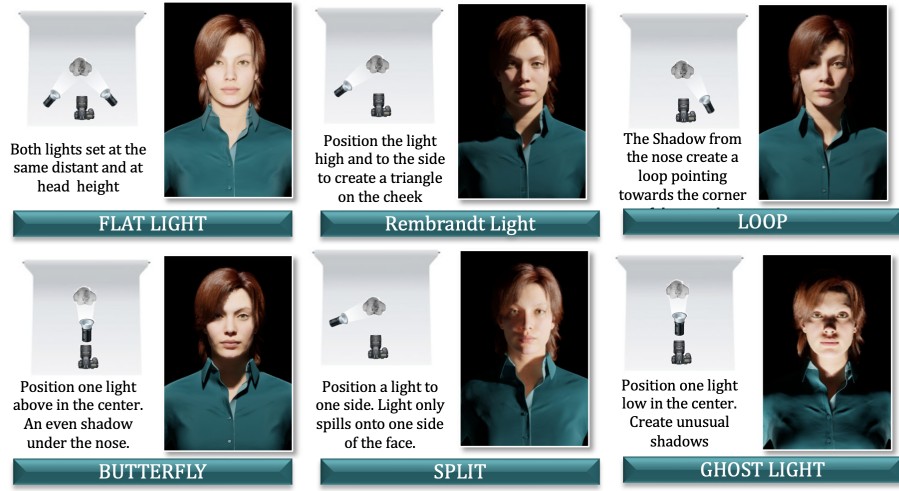

**Fig. 10.** *LumiHuman* offers a variety of basic elements that can be combined to form various types of portrait lighting, widely applicable to a range of tasks related to character lighting.

**Comment #4**

It would be important to highlight the key difference to ControlNet.

**Response:** Thanks. *LumiSculpt* is distinct from ControlNet in terms of its task, motivation, module design, model backbone, generated results, training objective and training data.

- **Task:** *LumiSculpt* is a specialized lighting control method designed for DiT based T2V models. Control-Net is a control method that focuses on image geometry (pose, depth map, canny, etc.) for U-Net based T2I models.
- **Motivation:** *LumiSculpt*'s motivation focuses on elements in videos that affect realism and aesthetics, i.e., lighting, and proposes a method to achieve coherent video generation with controllable lighting. ControlNet's motivation stems from the randomness in T2I diffusion models, hence it introduces a method for generating images with controllable geometry.
- **Module Design:** As shown in Fig. 11(d), *LumiSculpt* employs self-attention mechanisms as the lighting encoder and uses linear layers and latent weighting as condition injection mechanisms. ControlNet uses the U-Net Encoder to extract features and injects conditions by adding latents. These atomic components are commonly used and necessary for feature extraction and condition injection, which are not limited to a specific method.
- **Training Objective:** *LumiSculpt* tackles the core challenge of the entanglement of lighting and appearance. As shown in Fig. 11(c), *LumiSculpt* employs a dual-branch structure and an appearance-lighting disentanglement loss. ControlNet is trained with the diffusion noise prediction loss.
- **Training Data:** *LumiSculpt* utilizes video data with coherent inter-frame lighting changes, whereas ControlNet is based on independent images.
- **Backbone:** *LumiSculpt* is build upon DiT-based Open-Sora-Plan (Lab & etc., 2024), and ControlNet is designed for U-Net structured Stable Diffusion (Rombach et al., 2022).
- **Generated Results:** *LumiSculpt* generates coherent videos while ControlNet generates images.

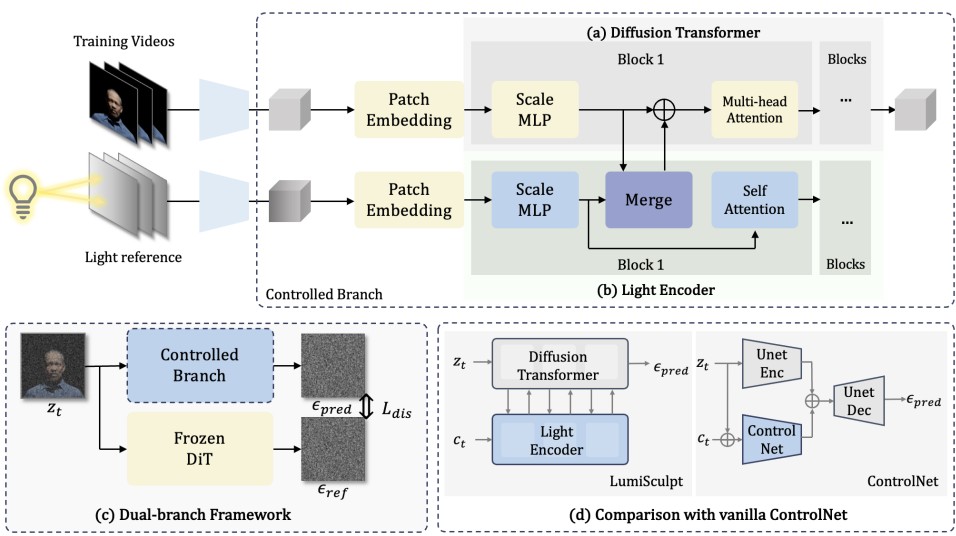

**Fig. 11.** Differences between *LumiSculpt* and ControlNet.

**Comment #5**

Now, it seems that the key difference is the dual-branch predictions, although the effectiveness of this idea is questionable based on the ablation. Furthermore, the proposed disentanglement loss is not well-motivated. The key assumption is that the latents reflect the appearance. However, the latents contain geometric, material, and also lighting features, thus not being disentangled.

**Response:** The dual-branch framework is proposed to address the core challenge of the entanglement of illumination and appearance. The proposed disentanglement loss is designed with the motivation for forcing the appearance distribution follow the backbone model, thus achieve disentanglement of appearance and lighting. **Specifically**, the $\mathcal{L}_{dis}$ calculates the mean and variance of each channel of the latent features, i.e. distributional differences between two latents without considering geometric features. This method of appearance disentanglement has been proven effective in a series of style transfer tasks (Huang & Belongie, 2017; Johnson et al., 2016). As shown in Fig. 12, without $\mathcal{L}_{dis}$, the background would overfit to black.

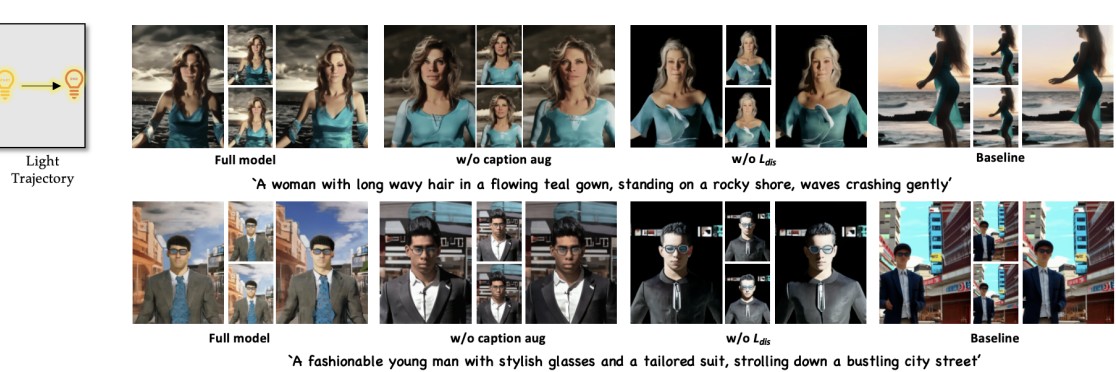

**Fig. 12.** Ablation results of $\mathcal{L}_{dis}$.

**Comment #6**

It would be great to show the diversity of the generated samples - more samples with the same conditioning.

**Response:** Thanks for the suggestion. As shown in Fig. 13, we present more results with the same prompt.

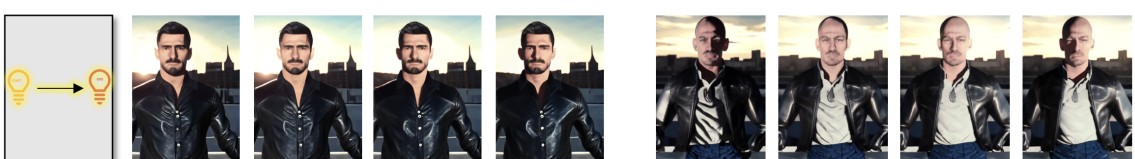

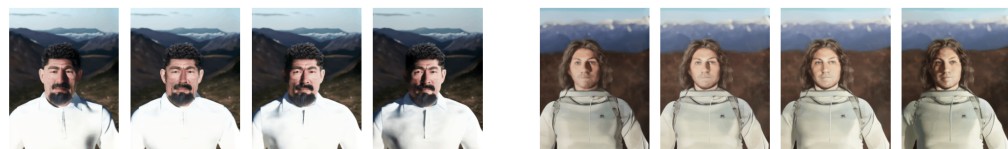

**Fig. 13.** *LumiSculpt* results with same prompt.

## Comment #7

Additional baseline comparisons would be important. Although the method uses the T2V models for light editing, the resulting videos are static, making it fair to compare against T2I models. Such comparisons could also give interesting insights about the lighting priors of T2I and T2V models.

**Response:** Thanks. The only appropriate open-source light control **T2I** methods is IC-Light. Existing relighting methods, such as Relightful Harmonization (Ren et al., 2024), target on **harmonizing** the lighting of a given foreground image and a background image. Our method achieves controllable lighting for T2V generation, where both characters and backgrounds are specified by text prompts. Therefore, relighting methods are not applicable to our task.

## Comment #8

The key contribution is not clear. Based on the title and abstract it is LumiSculpt, based on the intro (L.087 - Additionally...) it is the dataset LumiHuman.

**Response:** Thanks. We will revise the manuscript to avoid confusion. Both the dataset and methods are integral contributions of our work, which are **equally important**. Since we introduce a new task, it requires collecting suitable training data from scratch. The proposed *LumiHuman* dataset consists of videos showcasing varied and controllable lighting changes. Additionally, our model, *LumiSculpt*, is specifically designed for this task. The core contribution of *LumiSculpt* is achieving temporally stable light control through a DiT based generative model. **In conclusion**, the allocation of contributions in this work is similar to previous works like IC-Light (Zhang et al., 2024) and Relightful Harmonization (Ren et al., 2024), where the dataset and the method are equally significant.

## Comment #9

Recent T2I lighting control methods, such as LightIt could be discussed.

**Response:** Thanks for for introducing LightIt (Kocsis et al., 2024). We will cite this work and highlight the differences between LightIt and our approach. Specifically, LightIt is an image-guided (I2I) method for image relighting which requires additional estimated shading and normals. Our method, in contrast, is text-guided (T2V) and requires only text and target lighting conditions to achieve video lighting control. These differences provide valuable insights for our method design.

## Comment #10

It might be better to narrow the title, reflecting that the domain is human portraits.

**Response:** Thanks. *LumiSculpt* is **not restricted** to humans, we have experimented with some animal cases and also achieved stable lighting control effects, as shown in Fig. 14. It shows that *LumiSculpt* enables the model to learn about lighting priors and extend this knowledge to non-human objects.

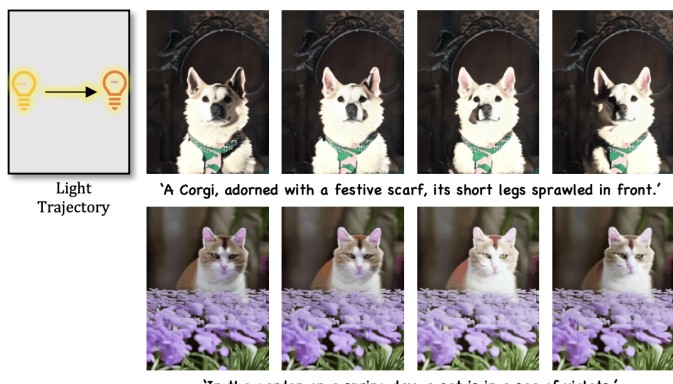

Light Trajectory

'A Corgi, adorned with a festive scarf, its short legs sprawled in front.'

'In the garden on a spring day, a cat is in a sea of violets.'

**Fig. 14.** *LumiSculpt* results with non-human objects.

---

**Comment #11**

What is the reason that the generated samples have a very similar geometry and appearance as IC Light, but highly different to Open-Sora, although the proposed method uses Open-Sora.

**Response:** This issue arises from our experimental settings. The foreground image fed to IC-Light is generated by *LumiSculpt*, as IC-Light is a relighting method that focuses on generating backgrounds and the overall lighting harmony. In contract, Open-Sora results are generated from random noise. It is worth noting that *LumiSculpt* is a fully functional and comprehensive T2V generative model designed to create controllable videos with lighting effects beyond relighting.

---

**Comment #12**

Could you please give a bit more details, how exactly are the augmented captions used? If I understand it correctly, the goal with those is to give additional noise to the model to avoid overfitting.

**Response:** The goal of the augmented captions is to provide regularization samples to the model to avoid overfitting. The regularization samples are latents of the same character against different backgrounds. Specifically, during training, the augmented captions serve as textual conditions into the dual-branch models. These captions can guide the frozen branch to produce latents for the same character against different backgrounds, which act as regularization samples providing strong appearance constraints for the $\mathcal{L}_{dis}$. This drives the Controlled Branch to generate richer backgrounds instead of only black backgrounds.

---

**Comment #13**

The results look oversaturated, what can be the reason for that?

**Response:** We are unsure which specific case the reviewer refers to regarding oversaturated. While some color deviations might occur due to the VAE and the pretrained backbone, overall, we think the results align well with standard aesthetic expectations. We employ the commonly used FID (Seitzer, 2020) score to assess the realism of the generated results for both *LumiSculpt* and Open-Sora (Lab & etc., 2024) within the FFHQ (Karras et al., 2019) dataset. As shown in Table VI, the FID score of *LumiSculpt* is better, demonstrating its ability to generate realistic videos.

Table VI: FID of *LumiSculpt* and Open-Sora using the FFHQ (Karras et al., 2019) dataset.

| Method | Open-Sora | LumiSculpt |
|---|---|---|
| FID $\downarrow$ | 35.7 | **33.0** |

# Referee: #4 LUeN

**Comment #1**

How diverse the MetaHuman dataset is since it is only contains 65 individuals.

**Response:**

The diversity of *LumiHuman* mainly lies in the variety of light trajectories rather than the individuals, leveraging varied lighting data to facilitate the model's learning of illumination rather than human appearance. Specifically, as shown in Tab. VII, compared to other lighting datasets Openillumination (Liu et al., 2024) and Deep Portrait Relighting (DPR) dataset (Zhou et al., 2019) (generated from face image dataset Celeb-A (Liu et al., 2015)), *LumiHuman* outperforms in light positions, light movements and number of images, which demonstrates the diversity of *LumiHuman*. Moreover, our *LumiHuman* of 65 human identities is sufficient for training *LumiSculpt*, which is supported by extensive qualitative and quantitative experiments. Fig. 15 shows real samples of human individuals in *LumiHuman*.

Table VII: Comparison of other lighting-related datasets.

| Dataset | Synthesis | Light Positions | Light Movement | Number of Images | Subject | Resolutions |
|---|---|---|---|---|---|---|
| DPR | 2D | 7 | None | 138K | - | $1024 \times 1024$ |
| Openillumination | Light Stage | 142 | None | 108K | 64 objects | $3000 \times 4096$ |
| LumiHuman | 3D | **35,937** | **>3K** | **2.3M** | **65 indivisuals** | $1024 \times 1024$ |

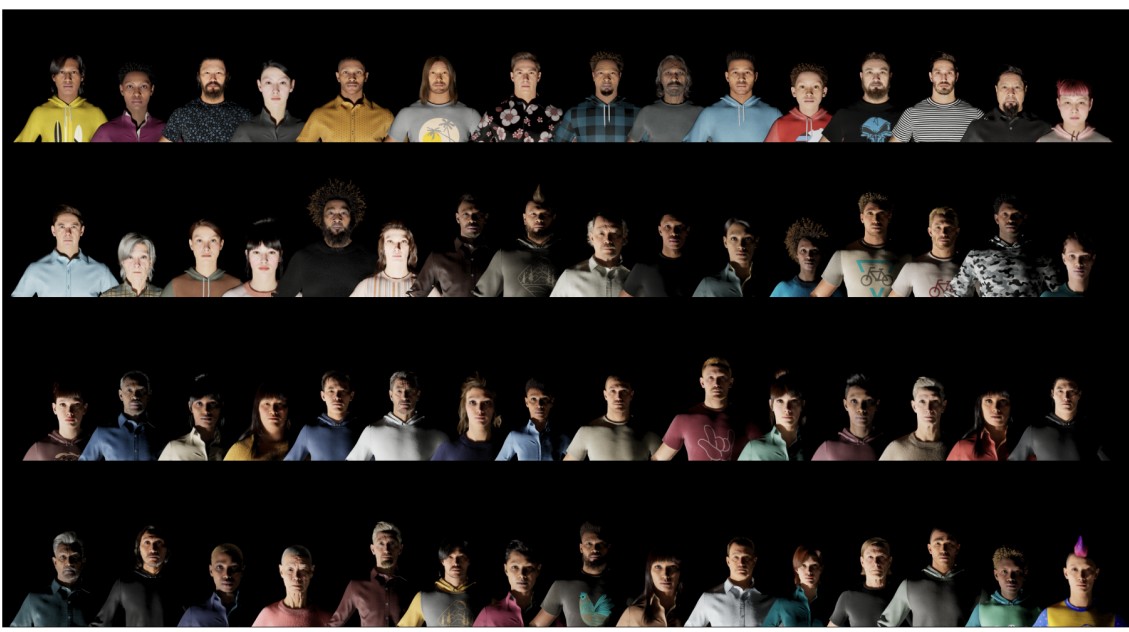

**Fig. 15.** Real samples in *LumiHuman*.

**Comment #2**

How accurate your caption could describe the lighting since lighting caption is a very unique task that current LLM model is not doing well. From the results, I didn't see any caption related to lighting.

**Response:** The caption only provides a supplementary semantic condition, such as background, character details, *etc*, and the precision of light control is guided by the input lighting reference video. Each frame in *LumiHuman* is paired with a lighting reference, allowing the descriptions of the lighting to be added to the captions, without relying on a Large Language Model (LLM). As commented by the reviewer, determining lighting remains a challenge for LLMs, and even for humans, since lighting itself is inherently difficult to describe in language. In contrast, the lighting reference video captures accurate lighting conditions, which serves as input and is easily interpreted by diffusion models.

**Comment #3**

Since the model is trained on synthetic rendered images, the results are far from photo-realistic and most of the results from the teaser images are 'fake' portrait with unrealistic facial texture.

**Response:** Thanks. Synthetic data does not compromise the model's generalization. During training, *LumiSculpt* also employ various strategies to mitigate overfitting, ensuring that our light control module primarily learns the patterns of light variation rather than the appearance or content of the characters. We employ the commonly used FID (Seitzer, 2020) score to assess the realism of the generated results for both *LumiSculpt* and Open-Sora (Lab & etc., 2024) within the FFHQ (Karras et al., 2019) dataset. As shown in Table VIII, the FID score of *LumiSculpt* is better, demonstrating its ability to generate realistic videos.

Table VIII: FID of *LumiSculpt* and Open-Sora using the FFHQ (Karras et al., 2019) dataset

| Method | Open-Sora | LumiSculpt |
|---|---|---|
| FID ↓ | 35.7 | **33.0** |

**Comment #4**

It is not clear how authors control the lighting intensity.

**When constructing *LumiHuman***, the light source distance varies in $50cm \sim 210cm$, which can create a noticeable effect of light intensity transitioning on the character's face. During **inference**, light intensity can be freely controlled using a user-specified lighting reference video. The light intensity of lighting reference video is changed by the distance between the light source and the illuminated subject. During model **training**, *LumiSculpt* can learn the mapping between the reference lighting intensity and the visual effects on the character's face from paired training data.

**Comment #5**

IC-light has much better photo-realistic results compared with your methods. And what's the advantage of authors method ?

**Response:** We kindly invite the reviewer to revisit our comparison results in Fig. 16 and the supplemented video. IC-Light fails to achieve stable lighting control in videos, as it is an image-based relighting method. It results in inconsistent lighitng across frames, with significant variations in both the subject and background across frames. It is worth noting that *LumiSculpt* is a fully functional and comprehensive T2V generative model designed to create controllable videos with lighting effects. While IC-Light does requires a portrait as the foreground input. Regarding photo-realistic, we shown the FID results in Response 3. Our method shows

fairly photo-realistic results.

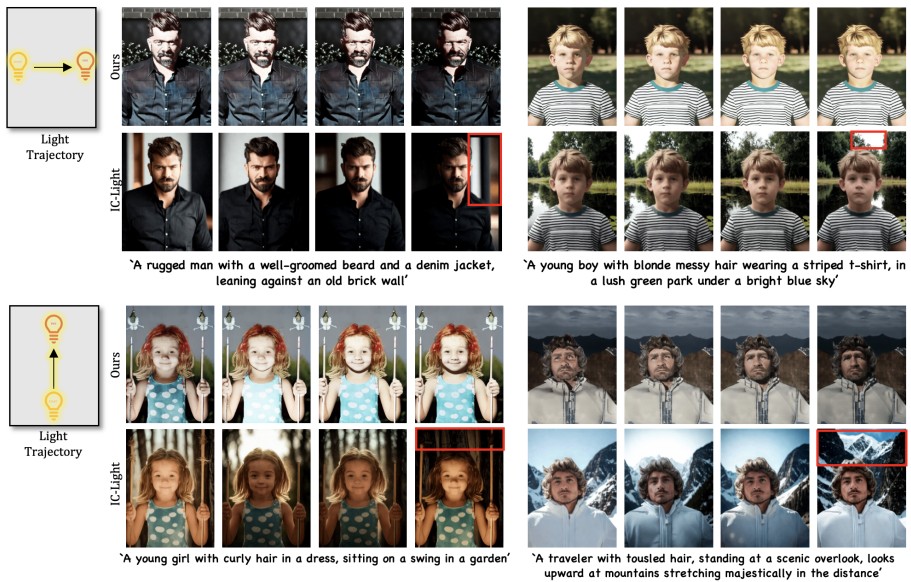

**Fig. 16.** Comparison results with state-of-the-art methods IC-Light (Zhang et al., 2024).

---

**Comment #6**

It seems that authors only show white/black lighting but not color lighting which ICnet could do.

---

**Response:** At present, no T2V generation methods are capable of controlling lighting, which is our primary objective. Modifying the color of the light is beyond the scope of our current work, which we plan to explore in future work.

---

**Comment #7**

Regarding model, I don't see any difference between yours and controlnet besides it is a video version.

---

**Response:** Thanks. The model of *LumiSculpt* is distinct from ControlNet in terms of its module design, backbone and training objective.

- **Module Design:** As shown in Fig. 17(d), *LumiSculpt* employs self-attention mechanisms as the lighting encoder and uses linear layers and latent weighting as condition injection mechanisms. ControlNet uses the U-Net Encoder to extract features and injects conditions by adding latents. These atomic components are commonly used and necessary for feature extraction and condition injection, which are not limited to a specific method.
- **Backbone:** *LumiSculpt* is build upon DiT-based Open-Sora-Plan (Lab & etc., 2024), and ControlNet is designed for U-Net structured Stable Diffusion (Rombach et al., 2022).
- **Training Objective:** *LumiSculpt* tackles the core challenge of the entanglement of lighting and appearance. As shown in Fig. 17(c), *LumiSculpt* employs a dual-branch structure and an appearance-lighting disentanglement loss. ControlNet is trained with the diffusion noise prediction loss.

We implement ControlNet to video lighting control by training with frames in *LumiHuman* and generating image sequence as video. The comparison results are shown in Fig. 18 and Tab. IX. ControlNet struggles to achieve lighting control, generating images with random lighting. This validates the effectiveness of our model structure and training methodology.

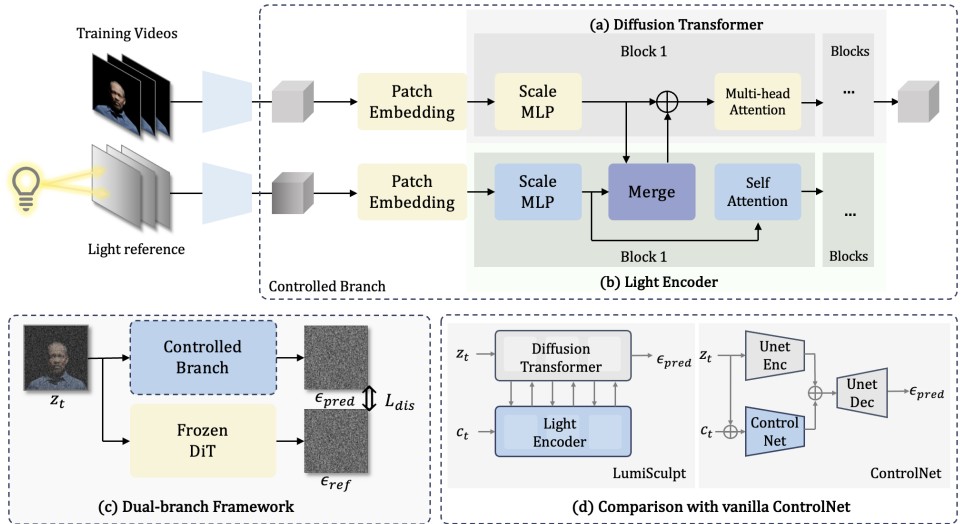

**Fig. 17.** Differences between *LumiSculpt* and ControlNet.

Table IX: Quantitative experimental results and ablation study results. The best results are marked as **bold**.

| Method | Consistency | | Lighting Accuracy | | Quality |
|---|---|---|---|---|---|
| | **CLIP↑** | **LPIPS↓** | **Direction↓** | **Brightness↑** | **CLIP↑** |
| Open-Sora | 0.9845 | 1.3503 | 0.4542 | 0.8229 | 0.3182 |
| IC-Light | 0.9703 | 2.5329 | 0.5264 | 0.8632 | 0.3145 |
| ControlNet | 0.8081 | 5.9324 | 0.5500 | 0.8032 | 0.3440 |
| Ours | **0.9951** | **1.1312** | **0.3500** | **0.8779** | **0.3597** |

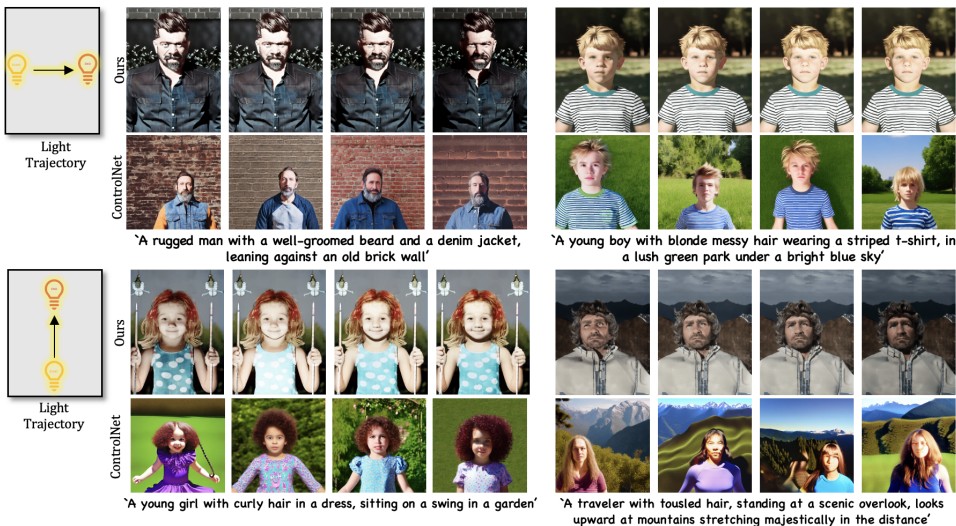

**Fig. 18.** Comparison results with state-of-the-art methods ControlNet (Zhang et al., 2023).

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
