# OpenReview forum: "LumiSculpt: A Consistency Lighting Control Network for Video Generation"
_ICLR.cc/2025/Conference — Submitted to ICLR 2025_

### Official Review · Reviewer_LUeN · 2024-10-31

**Soundness:** 2
**Presentation:** 2
**Contribution:** 2
**Rating:** 3
**Confidence:** 5

**Summary:**

This paper introduces LumiSculpt, a novel model that provides precise and consistent lighting control in text-to-video (T2V) generation. LumiSculpt. This allowing users to input custom lighting reference image sequences. The model includes a plug-and-play module that enables adjustable control over lighting intensity, position, and movement in latent video diffusion models, leveraging the advanced DiT (Denoising Transformer) backbone.

To train LumiSculpt effectively and address the lack of sufficient lighting data, the authors developed LumiHuman, a new, lightweight dataset specifically for portrait lighting in images and videos. Experiments demonstrate LumiSculpt’s ability to achieve precise, high-quality lighting control in video generation, marking a significant advancement in T2V lighting manipulation.

**Strengths:**

I felt the biggest pros of this paper is that authors explore how to propsoe a  LumiHuman dataset that could benefits current research community. But authors didn't mention whether they will release the dataset.

**Weaknesses:**

1. My point is that authors biggest contributions are the relighting dataset for video. However, the discussion on these dataset is so limited and there are a lot of details missing on the dataset.
For example: 1.1 How diverse the MetaHuman dataset is since it is only contains 65 individuals.
1.2 how accurate your caption could describe the lighting since  lighting caption is a very unique task that current LLM model is not doing well.
1.3 From the results, I didn't see any caption related to lighting.


2. Since the model is trained on synthetic rendered images, the results are far from photo realism and most of the results from the teaser images are 'fake' portrait with unrealistic facial texture.

3. It is not clear how authors control the lighting intensity.


4. IC-light has much better photo-realistic results compared with your methods. And what's the advantage of authors method ?


5.  It seems that authors only show white/black lighting but not color lighting which ICnet could do.


6. Regarding model, I don't see any difference between yours and controlnet besides it is a video version.

**Questions:**

See weakness.

**Details Of Ethics Concerns:**

It is trained on small amount of individuals with rendered synthetic dataset. Not sure how robust it is.

---

> ### Author Response · Authors · 2024-11-24
> **Thank you for your valuable feedbacks. Response(1)**
>
> We sincerely appreciate **Reviewer #4 LUeN** for acknowledging our work *"benefits current research community"*.
> We have re-uploaded our supplementary materials, which include the complete responses (at .zip/LumiSculpt_response_letter.pdf) along with the relevant figures and tables. The response letter is also contained in the main paper, after page 12. Below, we have addressed each question in detail and hope to clarify any concerns.
>
> ---
>
> **Comment #1 How diverse is the MetaHuman dataset since it only contains 65 individuals?**
>
> The diversity of LumiHuman mainly lies in the variety of light trajectories rather than the individuals, leveraging varied lighting data to facilitate the model’s learning of illumination rather than human appearance. Specifically, as shown in the table, compared to other lighting datasets Openillumination[1] and Deep Portrait Relighting (DPR) dataset[2] (generated from face image dataset Celeb-A[3]), LumiHuman outperforms in light positions, light movements, and number of images, which demonstrates the diversity of LumiHuman. Moreover, our LumiHuman of 65 human identities is sufficient for training LumiSculpt, which is supported by extensive qualitative and quantitative experiments.
>
> | Dataset         | Synthesis | Light Positions | Light Movement | Number of Images | Subject           | Resolutions    |
> |-----------------|-----------|----------------|----------------|-----------------|-----------------|--------------|
> | DPR            | 2D        | 7             | None           | 138K            | -               | $1024\times1024$ |
> | Openillumination | Light Stage | 142           | None           | 108K            | 64 objects       | $3000\times4096$ |
> | LumiHuman       | 3D        | **35,937**     | **>3K**        | **2.3M**        | **65 individuals** | $1024\times1024$ |
>
> [1] OpenIllumination: A Multi-Illumination Dataset for Inverse Rendering Evaluation on Real Objects. NeurIPS 2024
>
> [2] Deep Single Portrait Image Relighting. ICCV 2019
>
> [3] Deep Learning Face Attributes in the Wild. ICCV 2015
>
>
> ---
>
> **Comment \#2 How accurate your caption could describe the lighting since lighting caption is a very unique task that current LLM model is not doing well.
> From the results, I didn't see any caption related to lighting.**
>
> The caption only provides a supplementary semantic condition, such as background, character details, *etc*, and the precision of light control is guided by the input lighting reference video.
> Each frame in LumiHuman is paired with a lighting reference, allowing the descriptions of the lighting to be added to the captions, without relying on a Large Language Model (LLM).
> As commented by the reviewer, determining lighting remains a challenge for LLMs, and even for humans, since lighting itself is inherently difficult to describe in language.
> In contrast, the lighting reference video captures accurate lighting conditions, which serves as input and is easily interpreted by diffusion models.
>
>
> ---
>
> **Comment \#3 Since the model is trained on synthetic rendered images, the results are far from photo-realistic and most of the results from the teaser images are 'fake' portrait with unrealistic facial texture.**
>
> Thanks. Synthetic data does not compromise the model's generalization. During training, LumiSculpt also employ various strategies to mitigate overfitting, ensuring that our light control module primarily learns the patterns of light variation rather than the appearance or content of the characters.
> We employ the commonly used FID score to assess the realism of the generated results for both LumiSculpt and Open-Sora within the FFHQ[1] dataset.
> As shown in the table, the FID score of LumiSculpt is better, demonstrating its ability to generate realistic videos.
>
> | Method | Open-Sora | LumiSculpt |
> |--------|-----------|-----------|
> | FID ↓  | 35.7      | **33.0**  |
>
> [1] A Style-Based Generator Architecture for Generative Adversarial Networks. CVPR 2019
>
> ---
>
> **Comment \#4 It is not clear how authors control the lighting intensity.**
>
> **When constructing LumiHuman**, the light source distance varies from 50 cm to 210 cm, which can create a noticeable effect of light intensity transitioning on the character's face.
> During **inference**, light intensity can be freely controlled using a user-specified lighting reference video. The light intensity of the lighting reference video is changed by the distance between the light source and the illuminated subject.
> During **training**, LumiSculpt can learn the mapping between the reference lighting intensity and the visual effects on the character's face from paired training data.

---

> ### Author Response · Authors · 2024-11-24
> **Response(2)**
>
> **Comment \#5 IC-light has much better photo-realistic results compared with your methods. And what's the advantage of authors method ?**
>
> We kindly invite the reviewer to revisit our comparison results in the response letter and the supplemented video. IC-Light fails to achieve stable lighting control in videos, as it is an image-based relighting method. It results in inconsistent lighitng across frames, with significant variations in both the subject and background across frames.
> It is worth noting that LumiSculpt is a fully functional and comprehensive T2V generative model designed to create controllable videos with lighting effects.
> While IC-Light does requires a portrait as the foreground input.
> Regarding photo-realistic, we shown the FID results in Response(1). Our method shows fairly photo-realistic results.
>
> ---
>
> **Comment \#6 It seems that authors only show white/black lighting but not color lighting which ICnet could do.**
>
> At present, no T2V generation methods are capable of controlling lighting, which is our primary objective. Modifying the color of the light is beyond the scope of our current work, which we plan to explore in future work.
>
> ---
>
> **Comment \#7 Regarding model, I don't see any difference between yours and controlnet besides it is a video version.**
>
> Thanks.
> The model of LumiSculpt is distinct from ControlNet in terms of its module design, backbone and training objective.
> - **Module Design:**
>   LumiSculpt employs self-attention mechanisms as the lighting encoder and uses linear layers and latent weighting as condition injection mechanisms. ControlNet uses the U-Net Encoder to extract features and injects conditions by adding latents. These atomic components are commonly used and necessary for feature extraction and condition injection, which are not limited to a specific method.
> - **Backbone:**
>   LumiSculpt is built upon DiT-based Open-Sora-Plan, and ControlNet is designed for U-Net structured Stable Diffusion.
> - **Training Objective:**
>   LumiSculpt tackles the core challenge of the entanglement of lighting and appearance. LumiSculpt employs a dual-branch structure and an appearance-lighting disentanglement loss. ControlNet is trained with the diffusion noise prediction loss.
>
> We implement ControlNet to video lighting control by training with frames in LumiHuman and generating image sequence as video. The comparison results are shown in the table and the response letter. ControlNet struggles to achieve lighting control, generating images with random lighting. This validates the effectiveness of our model structure and training methodology.
>
> | Method     | Consistency (CLIP↑) | Lighting Accuracy (LPIPS↓) | Lighting Accuracy (Direction↓) | Lighting Accuracy (Brightness↑) | Quality (CLIP↑) |
> |------------|--------------------|--------------------------|----------------------------|------------------------------|-----------------|
> | Open-Sora  | 0.9845             | 1.3503                  | 0.4542                     | 0.8229                       | 0.3182          |
> | IC-Light   | 0.9703             | 2.5329                  | 0.5264                     | 0.8632                       | 0.3145          |
> | ControlNet | 0.8081             | 5.9324                  | 0.5500                     | 0.8032                       | 0.3440          |
> | **Ours**   | **0.9951**          | **1.1312**               | **0.3500**                  | **0.8779**                    | **0.3597**      |

---

### Official Review · Reviewer_wtpx · 2024-11-03

**Soundness:** 3
**Presentation:** 3
**Contribution:** 2
**Rating:** 5
**Confidence:** 5

**Summary:**

This paper proposes a control module for Text-2-Video models to enable lighting control. This is an extremely challenging task due to the lack of large-scale lighting datasets and the ambiguity between lighting and material properties. The key idea of the paper is to train a control network using a synthetic dataset, LumiHuman, consisting of 65 identities under various frontal illuminations defined over a uniform grid. The proposed dataset allows to combine the frames into various lighting trajectories. During training, a flat shading map is used as conditioning signal. The method shows impressive lighting control for static human portraits and is able to introduce cast shadows as well, which is really impressive.

**Strengths:**

* The proposed LumiHuman dataset can be valuable to the community. It is a great idea to place point lights close to each other to make it possible to compose frames into videos.
* The results look great, especially the cast shadows.
* The paper can provide interesting insights about the internal lighting prior of recent diffusion models.

**Weaknesses:**

I believe that the paper has multiple crucial flaws, which would require a major revision:
* **W1 -** The proposed dataset, which is one of the main contributions, seems to be a bit limited. The synthetic renderings could follow the usual light stage setup with full coverage, not just frontal lighting. Furthermore, it is not clear whether 65 identities can provide enough diversity. I believe that the main advantage of using a synthetic dataset is that it can be scaled. To verify this benefit, it would also be crucial to show that available real-world light-stage datasets cannot provide enough supervision to achieve such quality for lighting control.
* **W2 -** The novelty and effectiveness of the proposed pipeline, which is the second contribution, is not clear. It would be important to highlight the key difference to ControlNet. Now, it seems that the key difference is the dual-branch predictions, although the effectiveness of this idea is questionable based on the ablation. Furthermore, the proposed disentanglement loss is not well-motivated. The key assumption is that the latents reflect the appearance. However, the latents contain geometric, material, and also lighting features, thus not being disentangled.
* **W3 -** It would be great to show the diversity of the generated samples - more samples with the same conditioning.
* **W4 -** Additional baseline comparisons would be important. Although the method uses the T2V models for light editing, the resulting videos are static, making it fair to compare against T2I models. Such comparisons could also give interesting insights about the lighting priors of T2I and T2V models.
* **W5 -** The key contribution is not clear. Based on the title and abstract it is LumiSculpt, based on the intro (L.087 - Additionally...) it is the dataset LumiHuman.

Additional weaknesses
* **W6 -** Recent T2I lighting control methods, such as [LightIt](https://peter-kocsis.github.io/LightIt/) could be discussed.
* **W7 -** The notation on the pipeline figure is not clear. I would recommend naming the specific parameters instead of using "?". However, this might be due to some LaTex errors.
* **W8 -** Typo in L.204: "using a unreal engine" -> "using Unreal Engine [XYZ]"
* **W9 -** Typo in L.290: "an naive" -> "a naive"
* **W10 -** The formulation could be more precise. E.g., the lighting representation seems to be a flat shading map, but it is not clear.
* **W11 -** It might be better to narrow the title, reflecting that the domain is human portraits.
* **W12 -** The lighting is not entirely consistent with the results, e.g., in the first row of Fig. 1. the lighting seems to start from the bottom and move to the top right instead of moving only upwards as depicted in the trajectory.

**Questions:**

* **Q1 -** What is the reason that the generated samples have a very similar geometry and appearance as IC Light, but highly different to Open-Sora, although the proposed method uses Open-Sora.
* **Q2 -** Could you please give a bit more details, how exactly are the augmented captions used? If I understand it correctly, the goal with those is to give additional noise to the model to avoid overfitting.
* **Q3 -** The results look oversaturated, what can be the reason for that?

---

> ### Author Response · Authors · 2024-11-24
> **Thank you for your valuable feedbacks. Response(1)**
>
> We sincerely appreciate **Reviewer #3 wtpx** for acknowledging our work *"valuable to the community"*, *"results look great"* and *"provide interesting insights"*.
> We have re-uploaded our supplementary materials, which include the complete responses (at .zip/LumiSculpt_response_letter.pdf) along with the relevant figures and tables. The response letter is also contained in the main paper, after page 12. Below, we have addressed each question in detail and hope to clarify any concerns.
>
> ---
>
> **Comment \#1 The synthetic renderings could follow the usual light stage setup with full coverage, not just frontal lighting.**
>
> We sincerely appreciate your valuable suggestions regarding lighting settings. LumiHuman only includes light sources in front of the characters, because in an environment with point light sources, the light behind the characters would be blocked by the human body, resulting in a black image, or it appears as a near-white light spot, making it difficult to see the object. These phenomena exist in both generated data and real-world light-stage data.
>
> Our current light matrix is capable of creating rich light and shadow effects. LumiHuman provides over 30K lighting positions and over 3K lighting trajectories for each individual. These lighting positions can create light and shadow effects in **all areas** of the human face. We present the brightness distribution map of different regions of the human face in the response letter.
>
> ---
>
> **Comment \#2 Furthermore, it is not clear whether 65 identities can provide enough diversity. I believe that the main advantage of using a synthetic dataset is that it can be scaled.**
>
> Thanks for the comment. Our LumiHuman of 65 human identities can provide sufficient diversity to train LumiSculpt, which is supported by extensive qualitative and quantitative experiments. The scalability of synthetic data lies in the ability to construct diverse light trajectories, leveraging varied lighting data to facilitate the model's learning of illumination harmonization.
>
> As shown in the table, compared to other lighting datasets Openillumination[1] and Deep Portrait Relighting (DPR) dataset[2] (generated from face image dataset Celeb-A[3]), LumiHuman outperforms in light positions, light movements, and number of images.
>
> | Dataset             | Synthesis | Light Positions | Light Movement | Number of Images | Subject        | Resolutions      |
> |---------------------|-----------|----------------|---------------|-----------------|---------------|----------------|
> | DPR                | 2D        | 7             | None          | 138K            | -             | 1024×1024      |
> | Openillumination   | Light Stage | 142         | None          | 108K            | 64 objects     | 3000×4096      |
> | LumiHuman           | 3D        | **35,937**     | **>3K**       | **2.3M**        | **65 individuals** | 1024×1024 |
>
> [1] OpenIllumination: A Multi-Illumination Dataset for Inverse Rendering Evaluation on Real Objects. NeurIPS 2024
>
> [2] Deep Single Portrait Image Relighting. ICCV 2019
>
> [3] Deep Learning Face Attributes in the Wild. ICCV 2015
>
> ---
>
> **Comment \#3 It would also be crucial to show that available real-world light-stage datasets cannot provide enough supervision to achieve such quality for lighting control.**
>
> Thanks for the valuable suggestion.
> **Firstly**, available public light-stage datasets, e.g., Openillumination[1], do not contain human subject data, and its One-Light-At-a-Time (OLAT) data comprises only 142 lighting positions, which is hard to achieve smooth changes in lighting. Relying solely on publicly light-stage datasets is insufficient for T2V model training.
> **Secondly**, real-world light-stage datasets rely on HDR maps that have a significant domain gap with T2I and T2V scenarios.
> **In summary**, LumiHuman provides a coordinated, large spatial range of light sources, enabling users to freely combine the types of lighting they require.
>
> [1] OpenIllumination: A Multi-Illumination Dataset for Inverse Rendering Evaluation on Real Objects. NeurIPS 2024

---

> ### Author Response · Authors · 2024-11-24
> **Response(2)**
>
> **Comment \#4 It would be important to highlight the key difference to ControlNet.**
>
> LumiSculpt is distinct from ControlNet in terms of its task, motivation, module design, model backbone, generated results, training objective, and training data.
>
> - **Task:** LumiSculpt is a specialized lighting control method designed for DiT based T2V models.
>   ControlNet is a control method that focuses on image geometry (pose, depth map, canny, etc.) for U-Net based T2I models.
> - **Motivation:**
>   LumiSculpt's motivation focuses on elements in videos that affect realism and aesthetics, i.e., lighting, and proposes a method to achieve coherent video generation with controllable lighting.
>   ControlNet's motivation stems from the randomness in T2I diffusion models, hence it introduces a method for generating images with controllable geometry.
> - **Module Design:**
>   LumiSculpt employs self-attention mechanisms as the lighting encoder and uses linear layers and latent weighting as condition injection mechanisms. ControlNet uses the U-Net Encoder to extract features and injects conditions by adding latents.
>   These atomic components are commonly used and necessary for feature extraction and condition injection, which are not limited to a specific method.
> - **Training Objective:**
>   LumiSculpt tackles the core challenge of the entanglement of lighting and appearance. LumiSculpt employs a dual-branch structure and an appearance-lighting disentanglement loss.
>   ControlNet is trained with the diffusion noise prediction loss.
> - **Training Data:**
>   LumiSculpt utilizes video data with coherent inter-frame lighting changes, whereas ControlNet is based on independent images.
> - **Backbone:**
>   LumiSculpt is build upon DiT-based Open-Sora-Plan, and ControlNet is designed for U-Net structured Stable Diffusion.
> - **Generated Results:**
>   LumiSculpt generates coherent videos while ControlNet generates images.
>
> ---
>
> **Comment \#5 Now, it seems that the key difference is the dual-branch predictions, although the effectiveness of this idea is questionable based on the ablation.
> Furthermore, the proposed disentanglement loss is not well-motivated. The key assumption is that the latents reflect the appearance. However, the latents contain geometric, material, and also lighting features, thus not being disentangled.**
>
> The dual-branch framework is proposed to address the core challenge of the entanglement of illumination and appearance.
> The proposed disentanglement loss is designed with the motivation for forcing the appearance distribution follow the backbone model, thus achieve disentanglement of appearance and lighting.
> **Specifically**, the disentanglement loss calculates the mean and variance of each channel of the latent features, i.e., distributional differences between two latents without considering geometric features.
> This method of appearance disentanglement has been proven effective in a series of style transfer tasks[1,2].
>
> [1] Arbitrary style transfer in real-time with adaptive instance normalization. ICCV 2017
>
> [2] Perceptual losses for real-time style transfer and super-resolution. ECCV 2016
>
> ---
>
> **Comment \#6 It would be great to show the diversity of the generated samples - more samples with the same conditioning.**
>
> Thanks for the suggestion. We present more results with the same prompt in the response letter. LumiSculpt can generate diverse appearance with same prompts.
>
> ---
>
> **Comment \#7 Additional baseline comparisons would be important. Although the method uses the T2V models for light editing, the resulting videos are static, making it fair to compare against T2I models. Such comparisons could also give interesting insights about the lighting priors of T2I and T2V models.**
>
> Thanks. The only appropriate open-source light control **T2I** methods is IC-Light.
> Existing relighting methods, such as Relightful Harmonization[1], target on **harmonizing** the lighting of a given foreground image and a background image.
> Our method achieves controllable lighting for T2V generation, where both characters and backgrounds are specified by text prompts.
> Therefore, relighting methods are not applicable to our task.
>
> [1] Relightful Harmonization: Lighting-aware Portrait Background Replacement. CVPR 2024

---

> ### Author Response · Authors · 2024-11-24
> **Response(3)**
>
> **Comment \#8 The key contribution is not clear. Based on the title and abstract it is LumiSculpt, based on the intro (L.087 - Additionally...) it is the dataset LumiHuman.**
>
> Thanks. We will revise the manuscript to avoid confusion.
> Both the dataset and methods are integral contributions of our work, which are **equally important**.
> Since we introduce a new task, it requires collecting suitable training data from scratch. The proposed LumiHuman dataset consists of videos showcasing varied and controllable lighting changes.
> Additionally, our model, LumiSculpt, is specifically designed for this task. The core contribution of LumiSculpt is achieving temporally stable light control through a DiT based generative model.
> **In conclusion**, the allocation of contributions in this work is similar to previous works like IC-Light[1] and Relightful Harmonization[2], where the dataset and the method are equally significant.
>
> [1] Scaling In-the-Wild Training for Diffusion-based Illumination Harmonization and Editing by Imposing Consistent Light Transport
>
> [2] Relightful Harmonization: Lighting-aware Portrait Background Replacement. CVPR 2024
>
> ---
>
> **Comment \#9 Recent T2I lighting control methods, such as LightIt could be discussed.**
>
> Thanks for introducing LightIt[1]. We will cite this work and highlight the differences between LightIt and our approach. Specifically, LightIt is an image-guided (I2I) method for image relighting which requires additional estimated shading and normals.
> Our method, in contrast, is text-guided (T2V) and requires only text and target lighting conditions to achieve video lighting control.
> These differences provide valuable insights for our method design.
>
> [1] LightIt: Illumination Modeling and Control for Diffusion Models. CVPR 2024
>
> ---
>
> **Comment \#10 It might be better to narrow the title, reflecting that the domain is human portraits.**
>
> Thanks. LumiSculpt is **not restricted** to humans. We have experimented with some animal cases and also achieved stable lighting control effects. The results are shown in the response letter.
>
> ---
>
> **Comment \#11 What is the reason that the generated samples have a very similar geometry and appearance as IC Light, but highly different to Open-Sora, although the proposed method uses Open-Sora.**
>
> This issue arises from our experimental settings. The foreground image fed to IC-Light is generated by LumiSculpt, as IC-Light is a relighting method that focuses on generating backgrounds and the overall lighting harmony. In contrast, Open-Sora results are generated from random noise.
> It is worth noting that LumiSculpt is a fully functional and comprehensive T2V generative model designed to create controllable videos with lighting effects beyond relighting.
>
> ---
>
> **Comment #12 Could you please give a bit more details, how exactly are the augmented captions used? If I understand it correctly, the goal with those is to give additional noise to the model to avoid overfitting.**
>
> **Response:**
> The goal of the augmented captions is to provide regularization samples to the model to avoid overfitting. The regularization samples are latents of the same character against different backgrounds.
>
> Specifically, during training, the augmented captions serve as textual conditions into the dual-branch models. These captions can guide the frozen branch to produce latents for the same character against different backgrounds, which act as regularization samples providing strong appearance constraints for the disentanglement loss. This drives the Controlled Branch to generate richer backgrounds instead of only black backgrounds.
>
> ---
>
> **Comment #13 The results look oversaturated, what can be the reason for that?**
>
> We are unsure which specific case the reviewer refers to regarding oversaturated. While some color deviations might occur due to the VAE and the pretrained backbone, overall, we think the results align well with standard aesthetic expectations. We employ the commonly used FID score to assess the realism of the generated results for both LumiSculpt and Open-Sora within the FFHQ[1] dataset. As shown in the table, the FID score of LumiSculpt is better, demonstrating its ability to generate realistic videos.
>
> | Method | Open-Sora | LumiSculpt |
> |--------|-----------|-----------|
> | FID ↓  | 35.7      | **33.0**  |
>
> [1] A Style-Based Generator Architecture for Generative Adversarial Networks. CVPR 2019

---

> > ### Comment · Reviewer_wtpx · 2024-11-29
> >
> > I thank the authors for their thorough response! I really appreciate the authors extra effort and the additional details.
> >
> > Comment #2
> > I agree with reviewer LUeN that the current results do not  underline that 65 human identities can provide diverse enough supervision, although, I can see the point that the diffusion model already has a strong prior over humans, making it possible to generlize well even with such a small dataset.
> >
> > Comment #4
> > I believe that the shared comment of the reviewers is that the proposed method uses the key idea of ControlNet. I am referring to the idea of training a partial copy of the original model and use intermediate features to control the generation process. This approach is not limited to image generation and convolutional architecture, as ControlNets are already available for DiTs as well. Therefore, I believe that the key concept is identical, which undermines the novelty of the proposed architecture.
> >
> > Comment #5
> > I thank the authors for their description. However, I am still not convinced, how the disentanglement loss helps with disentangling appearance and lighting. First, I still believe that the latent features have geometric features as well, not just material and lighting. Second, Making regularizing the distributional difference between the trained and frozen model's features do not enforce any appearance decomposition, but rather ensures prior preservation. This also makes sense, helps to maintain backgound generation capabilities and gives better generalization, but does not ensure any disentanglement.
> >
> > Comment #11
> > The repeated argument of the authors that their task it lighting-controllable video generation feels a bit overstated. Since the proposed dataset consists only static humans, all the results show only static (or slightly moving) objects. This is the reason, why I mentioned comparisons to T2I models. Currently, I feel that the method rather utilizes the prior of T2V models for lighting-controllable image generation.
> >
> > Comment #13
> > As the reviewer MY7D also pointed out, I also refer to the foreground, e.g. third row of Fig.1. is highly oversatured.
> >
> >
> > Overall, my main concern is still the novelty of the method. I really appreciate the dataset and I believe it can be valuable to the community; thus, I increase my score. However, the method sits between T2I and T2V models and is highly similar ControlNet; therefore, I am stil leaning to reject.

---

> > > ### Author Response · Authors · 2024-12-01
> > > **Thanks a lot for the suggestions and valuable comments!**
> > >
> > > Dear Reviewer wtpx,
> > >
> > > We appreciate your patient review of our response and the  decision to raise the score of our work.
> > > Thanks a lot for the valuable suggestions.
> > >
> > > Best regards,
> > >
> > > LumiSculpt authors

---

### Official Review · Reviewer_MY7D · 2024-11-03

**Soundness:** 3
**Presentation:** 2
**Contribution:** 3
**Rating:** 6
**Confidence:** 4

**Summary:**

The paper propose a approach to precise lighting control in text-to-video (T2V) generation. The authors design a model named LumiSculpt, which introduces plug-and-play lighting control to enhance video generation with adjustable lighting intensity, direction, and trajectory, all controlled through textual inputs. LumiSculpt uses a dual-branch framework with a frozen branch to ensure diversity in appearances while maintaining lighting control.  Besides, the authors introduce a specialized dataset, LumiHuman, containing diverse portrait lighting data generated via Unreal Engine, which supports various lighting trajectories.  The model’s performance is evaluated against state-of-the-art methods, Open-Sora and IC-Light, demonstrating its superiority in lighting accuracy, inter-frame consistency, and semantic alignment with text descriptions.

**Strengths:**

+ The model design is novel. The introduction of an additional lighting embedding in the diffusion process is reasonable and easy for model to control the lighting. The loss design is cool. It is based on the concept of style transfer but in the latent space. This loss controls the lighting intensity and direction while preserving the appearance.
+ This paper prepares a dataset to enhance lighting control in video generation. Although the sample size is limited, the pipeline design can provide valuable insights for future work.

**Weaknesses:**

- This algorithm seems more suitable for image generation, as I did not observe any specific design tailored for video tasks. Video generation is merely an extension of the algorithm’s application.
- In the comparisons, the authors use images generated by the network as the foreground. Does this imply that, limited by the synthetic data used during training, the algorithm may not generalize well to real-world scenes? I also noticed unnatural foreground (human) generation results in the video demo.

**Questions:**

In addition to the weaknesses mentioned, I have the following concerns:
1. Can this dataset be open-sourced to ensure reproducibility for future work?
2. I find the caption augmentation section somewhat unclear. Is it simply replacing captions, or does it involve corresponding changes in the image background as well?

---

> ### Author Response · Authors · 2024-11-24
> **Thank you for your valuable feedbacks.**
>
> We sincerely appreciate **Reviewer #2 MY7D** for acknowledging *"The model design is novel"*, *"The loss design is cool"* and *"provide valuable insights for future work"*.
> We have re-uploaded our supplementary materials, which include the complete responses (at .zip/LumiSculpt_response_letter.pdf) along with the relevant figures and tables. The response letter is also contained in the main paper, after page 12. Below, we have addressed each question in detail and hope to clarify any concerns.
>
> ---
>
> **Comment \#1 This algorithm seems more suitable for image generation, as I did not observe any specific design tailored for video tasks. Video generation is merely an extension of the algorithm’s application.**
>
> Thanks for the comment.
> **Firstly**, LumiSculpt incorporated 3D attention specifically designed for temporal modeling in videos. All light injection modules in this work are built upon the backbone of the video diffusion generation model, ensuring consistent temporal modeling of light dynamics without compromising the model's original generative capabilities.
> **Secondly**, lighting control in image generation primarily focuses on harmonizing lighting between the background and the subject. When directly applied the image based method to video generation, it may result in severe temporal inconsistencies, as each frame may exhibit different visual content. In contrast, our approach demonstrates smooth and stable lighting across video frames, reflecting the effectiveness of our current design, which including conditional extraction and injection methods, for video generation.
>
> ---
>
> **Comment \#2 In the comparisons, the authors use images generated by the network as the foreground. Does this imply that, limited by the synthetic data used during training, the algorithm may not generalize well to real-world scenes? I also noticed unnatural foreground (human) generation results in the video demo.**
>
> Thanks for your suggestion. Synthetic data does not compromise the model's generalization. During training, LumiSculpt also employs various strategies to mitigate overfitting, ensuring that our light control module primarily learns the patterns of light variation rather than the appearance or content of the characters. We employ the commonly used FID score to assess the realism of the generated results for both LumiSculpt and Open-Sora within the FFHQ[1] dataset. As shown in the table, LumiSculpt achieves a better FID score, demonstrating its ability to generate realistic videos.
>
> | Method      | Open-Sora | LumiSculpt |
> |-------------|-----------|-----------|
> | FID ↓       | 35.7      | **33.0**  |
>
> LumiSculpt is a T2V method, aiming at generating lighting controllable videos by texts. Thus, the ability to generate both foreground and background with text is an advantage of LumiSculpt. IC-Light's goal is relighting, which involves harmonizing lighting between foreground and background images. Thus, IC-Light's foreground is generated by LumiSculpt because it needs a foreground image.
>
> [1] A Style-Based Generator Architecture for Generative Adversarial Networks. CVPR 2019
>
> ---
>
> **Comment \#3 Can this dataset be open-sourced to ensure reproducibility for future work?**
>
> Yes, it certainly will be open-sourced upon acceptance.
>
> ---
>
> **Comment \#4 I find the caption augmentation section somewhat unclear. Is it simply replacing captions, or does it involve corresponding changes in the image background as well?**
>
> It is replacing captions. During training, the augmented captions serve as textual conditions input into the dual-branch models. These captions can guide the frozen branch to produce latents for the same character against different backgrounds, which act as regularization samples providing strong appearance constraints for the disentanglement loss. This drives the Controlled Branch to generate richer backgrounds instead of only black backgrounds. The inclusion of augmented captions enhances the model's ability to generate diverse backgrounds and layouts.

---

### Official Review · Reviewer_rLXS · 2024-11-04

**Soundness:** 3
**Presentation:** 3
**Contribution:** 3
**Rating:** 5
**Confidence:** 5

**Summary:**

This paper proposes LumiSculpt to deal with the lighting control for video generation. Controlling lighting is a fundamental need for image and video synthesis tasks. This work can control the light intensity, position, and trajectory. Besides the proposed model, this work also constructs a dataset composed of portraits with different lighting conditions.

**Strengths:**

1. This work achieves precise control of the lighting of the portrait in the video generation tasks.
2. The plug-and-play design of the module makes this work flexible to be combined with other foundation models.
3. The LumiHuman will benefit the community if it can be made public available.

**Weaknesses:**

1. From what I understand, the LumiHuman is synthetic, which may limit the model's performance in real-world cases. I wonder if there can be a thorough evaluation of real-world cases. Meanwhile, there are only 65 individuals in the dataset, which may limit the model to generalize to new portraits.
2. The generated videos are not informative enough. The motion dynamics are not enough. I wonder if there are results where the portrait and background can move more vividly.

**Questions:**

More cases on real-world cases.

---

> ### Author Response · Authors · 2024-11-24
> **Thank you for your valuable feedbacks.**
>
> We sincerely appreciate **Reviewer #1 rLXS** for acknowledging our work *"achieves precise control of the lighting"*, *"flexible"* and *"benefit the community"*.
> We have re-uploaded our supplementary materials, which include the complete responses (at .zip/LumiSculpt_response_letter.pdf) along with the relevant figures and tables. The response letter is also contained in the main paper, after page 12. Below, we have addressed each question in detail and hope to clarify any concerns.
>
> ---
>
> **Comment \#1 LumiHuman is synthetic, which may limit the model's performance in real-world cases. I wonder if there can be a thorough evaluation of real-world cases. There are only 65 individuals in the dataset, which may limit the model to generalize to new portraits.**
>
> Thanks for your suggestion. Synthetic data does not compromise the model's generalization. During training, LumiSculpt employs various strategies to mitigate overfitting, ensuring that the light control module primarily learns the patterns of light variation rather than the appearance of the characters. To evaluate with real-world case, we employ the commonly used FID score to assess the photo-realism of both LumiSculpt and Open-Sora within the FFHQ[1] dataset. As shown in the table, LumiSculpt achieves a better FID score, demonstrating its ability to generate realistic videos.
>
> | Method      | Open-Sora | LumiSculpt |
> |-------------|-----------|-----------|
> | FID ↓       | 35.7      | **33.0**  |
>
> The 65 individuals are also not the limiting factor for model training. LumiSculpt learns lighting variation patterns and achieves generalization through diverse light trajectories constructed from synthetic data, rather than relying on human appearances.
>
> [1] A Style-Based Generator Architecture for Generative Adversarial Networks. CVPR 2019
>
> ---
>
> **Comment \#2 The generated videos are not informative enough. The motion dynamics are not enough. I wonder if there are results where the portrait and background can move more vividly.**
>
> Yes, with motion descriptions, LumiSculpt exhibits motion dynamics where the portrait and background can move more vividly. We shown results in the response letter.
> Actually, generating portrait and background with vivid dynamics is challenging for T2V models, and it is even harder to control both lighting and motion dynamics.
> Applying image-based lighting control methods (since there is no suitable video-based model available) cannot achieve inter-frame consistency.
> Therefore, LumiSculpt provides a novel solution for controllable video generation, particularly focused on lighting.

---

> > ### Comment · Reviewer_rLXS · 2024-11-27
> >
> > After reading your rebuttal, I still have concerns regarding the model's generalization ability to real-world cases. I would like to see some visual results, if time permits. The motion dynamics of the generated videos are still  not satisfactory enough, from my perspective. Moreover, some of the generated videos are over-exposured, which significantly degrades the reallism. Overall, the current results shown by the authors are not convincing enough nor ready enough to be accepted by this conference.

---

> > > ### Author Response · Authors · 2024-11-28
> > > **Thanks. More results including human and animals are added in response letter page 4, main paper page 17.**
> > >
> > > Dear Reviewer rLXS,
> > >
> > > Thank you for your suggestion.
> > > We have added more results with varying backgrounds, characters, and even **animals** on page 4 of the response letter and page 17 of the main paper. These demonstrate LumiSculpt's ability to generalize to real-world scenarios.
> > > Some results in the main paper that appear overexposed are a common occurrence when the light source is close to the person, similar to what happens in photography. To achieve a softer visual effect, the aforementioned new results utilized lighting from a longer distance.
> > >
> > > Best regards,
> > >
> > > LumiSculpt authors

---

### Author Response · Authors · 2024-11-24
**We would like to thank all the reviewers for their careful reading and insightful comments.**

We would like to thank all the reviewers for their careful reading and insightful comments on the manuscript. We sincerely appreciate reviewers for acknowledging:

- The proposed dataset will benefit the community(r1,r2,r3,r4).

- Our work achieves precise lighting control(r1,r3).

- The proposed module and loss design are novel and flexible(r1,r2).

We have re-uploaded our supplementary materials, which include the complete **response letter** (at .zip/LumiSculpt_response_letter.pdf) along with the relevant figures and tables. The response letter is also contained in the main paper, after page 12. Below, we have addressed each question in detail and hope to clarify any concerns.
We sincerely invite the reviewers to refer to these materials for a better reading experience. We hope that our response satisfactorily addresses your concerns.

We have addressed the common comments:

---

**[Common Concern 1] Statistics and Diversity of LumiHuman Dataset**

We introduce the LumiHuman dataset, a continuous lighting video dataset comprising over **195K** different videos (*i.e.*, **2.3 million** images). The resolution of each video is 1024×1024.
LumiHuman includes 65 diverse human subjects, 30K lighting positions, and over 3K lighting trajectories for each people.

LumiHuman of 65 human identities is sufficient for training LumiSculpt, which is supported by extensive qualitative and quantitative experiments. The scalability and diversity of synthetic data lies in the ability to construct diverse light trajectories, leveraging varied lighting data to facilitate the model's learning of illumination.

As shown in the table below, compared to
other lighting datasets Openillumination[1] and Deep Portrait Relighting (DPR) dataset[2] (generated from face image dataset Celeb-A[3]), LumiHuman outperforms in
light positions, light movements and number of images.

| Dataset             | Synthesis | Light Positions | Light Movement | Number of Images | Subject        | Resolutions      |
|---------------------|-----------|----------------|---------------|-----------------|---------------|----------------|
| DPR                | 2D        | 7             | None          | 138K            | -             | 1024×1024      |
| Openillumination   | Light Stage | 142         | None          | 108K            | 64 objects     | 3000×4096      |
| LumiHuman           | 3D        | **35,937**     | **>3K**       | **2.3M**        | **65 individuals** | 1024×1024 |

[1] OpenIllumination: A Multi-Illumination Dataset for Inverse Rendering Evaluation on Real Objects. NeurIPS 2024

[2] Deep Single Portrait Image Relighting. ICCV 2019

[3] Deep Learning Face Attributes in the Wild. ICCV 2015

---

> ### Comment · Reviewer_LUeN · 2024-11-24
> **Disagree on data diversity**
>
> “LumiHuman of 65 human identities is sufficient for training LumiSculpt, which is supported by extensive qualitative and quantitative experiments. The scalability and diversity of synthetic data lies in the ability to construct diverse light trajectories, leveraging varied lighting data to facilitate the model's learning of illumination.”
>
>
> I don't believe it is sufficient that 65 human identity are enough.   And I don't believe authors did comprehensive testing on that especially related to skin color.

---

> > ### Author Response · Authors · 2024-11-24
> > **Thanks.  LumiHuman assists the model in learning light effects on the human face, rather than the physical appearance.**
> >
> > We kindly invite the reviewer to refer to Fig.1 in our response letter. It shows the diversity of individuals in LumiHuman, including people of various skin colors.
> > The appearance of individuals is near infinite, and modeling human appearance is the task of the base model during T2I/T2V training, and it does not rely on LumiHuman.
> > The arrangement of human facial features follows certain regular patterns. LumiHuman assists the model in learning the visual effects caused by light on the human face, such as shadows, **rather than the physical appearance**.

---

### Author Response · Authors · 2024-11-24
**[Common Concern 2]**

**[Common Concern 2] Similar to ControlNet**

LumiSculpt is distinct from ControlNet in terms of its task, motivation, module design, training objective, training data, backbone, and generated results. A detailed explanation for each point is provided below:

- **Task:** LumiSculpt is a specialized lighting control method designed for DiT based T2V models.
  ControlNet is a control method that focuses on image geometry (pose, depth map, canny, etc.) for U-Net based T2I models.
- **Motivation:**
  LumiSculpt's motivation focuses on elements in videos that affect realism and aesthetics, i.e., lighting, and proposes a method to achieve coherent video generation with controllable lighting.
  ControlNet's motivation stems from the randomness in T2I diffusion models, hence it introduces a method for generating images with controllable geometry.
- **Module Design:**
  LumiSculpt employs self-attention mechanisms as the lighting encoder and uses linear layers and latent weighting as condition injection mechanisms. ControlNet uses the U-Net Encoder to extract features and injects conditions by adding latents.
  These atomic components are commonly used and necessary for feature extraction and condition injection, which are not limited to a specific method.
- **Training Objective:**
  LumiSculpt tackles the core challenge of the entanglement of lighting and appearance. LumiSculpt additionally employs a dual-branch structure and introduces an appearance-lighting disentanglement loss.
  ControlNet is trained with the diffusion noise prediction loss.
- **Training Data:**
  LumiSculpt utilizes video data with coherent inter-frame lighting changes, whereas ControlNet is based on independent images.
- **Backbone:**
  LumiSculpt is build upon DiT-based Open-Sora-Plan, and ControlNet is designed for U-Net structured Stable Diffusion.
- **Generated Results:**
  LumiSculpt generates coherent videos while ControlNet generates images.

We implement ControlNet to video lighting control by training with paired frames in LumiHuman and generating image sequence as video. The comparison results are shown in the table and in the response letter.

| Method      | Consistency (CLIP↑) | Lighting Accuracy (LPIPS↓) | Lighting Accuracy (Direction↓) | Lighting Accuracy (Brightness↑) | Quality (CLIP↑) |
|-------------|--------------------|--------------------------|-----------------------------|------------------------------|-----------------|
| Open-Sora   | 0.9845             | 1.3503                  | 0.4542                      | 0.8229                        | 0.3182          |
| IC-Light    | 0.9703             | 2.5329                  | 0.5264                      | 0.8632                        | 0.3145          |
| ControlNet  | 0.8081             | 5.9324                  | 0.5500                      | 0.8032                        | 0.3440          |
| **Ours**    | **0.9951**          | **1.1312**               | **0.3500**                   | **0.8779**                    | **0.3597**      |

The best results are marked as **bold**.

---

### Meta-Review · Area_Chair_7Lpj · 2024-12-09

**Metareview:**

The paper receives mostly negative scores from reviewers. While reviewer appreciate the simple yet effective design, they also found the result quality limited, the extension from images to videos unclear, and questioned the generalizability to real-world cases. The authors are encouraged to address these comments in the revised version.

**Additional Comments On Reviewer Discussion:**

During the discussion, reviewer MY7D agrees with other reviewers that the result quality is unnatural and the synthetic dataset is limited, and is ok with rejecting the paper.

---

### Decision · Program_Chairs · 2025-01-22

Reject